# FROM POINTS TO FUNCTIONS: INFINITE-DIMENSIONAL REPRESENTATIONS IN DIFFUSION MODELS

## ABSTRACT

Diffusion-based generative models learn to iteratively transfer unstructured noise to a complex target distribution as opposed to Generative Adversarial Networks (GANs) or the decoder of Variational Autoencoders (VAEs) which produce samples from the target distribution in a single step. Thus, in diffusion models every sample is naturally connected to a random trajectory which is a solution to a learned stochastic differential equation (SDE). Generative models are only concerned with the final state of this trajectory that delivers samples from the desired distribution. Abstreiter et al. (2021) showed that these stochastic trajectories can be seen as continuous filters that wash out information along the way. Consequently, it is reasonable to ask if there is an intermediate time step at which the preserved information is optimal for a given downstream task. In this work, we show that a combination of information content from different time steps gives a strictly better representation for the downstream task. We introduce an attention and recurrence based modules that "learn to mix" information content of various time-steps such that the resultant representation leads to superior performance in downstream tasks.

## 1 INTRODUCTION

A lot of the progress in Machine Learning hinges on learning good representations of the data, whether in supervised or unsupervised fashion. Typically in the absence of label information, learning a good representation is often guided by reconstruction of the input, as is the case with autoencoders and generative models like variational autoencoders (Vincent et al., 2010; Kingma & Welling, 2013; Rezende et al., 2014); or by some notion of invariance to certain transformations like in Contrastive Learning and similar approaches (Chen et al., 2020b;d; Grill et al., 2020). In this work, we analyze a novel way of representation learning which was introduced in Abstreiter et al. (2021) with a denoising objective using diffusion based models to obtain unbounded representations.

Diffusion-based models (Sohl-Dickstein et al., 2015; Song et al., 2020; 2021; Sajjadi et al., 2018; Niu et al., 2020; Cai et al., 2020; Chen et al., 2020a; Saremi et al., 2018; Dhariwal & Nichol, 2021; Luhman & Luhman, 2021; Ho et al., 2021; Mehrjou et al., 2017; Nichol & Dhariwal, 2021) are generative models that leverage step-wise perturbations to the samples of the data distribution (eg. CIFAR10), modeled via a Stochastic Differential Equation (SDE), until convergence to an unstructured distribution (eg. $\mathcal{N}(\mathbf{0}, \mathbf{I})$) called, in this context, the prior distribution. In contrast to this diffusion process, a "score model" is learned to approximate the reverse process that iteratively converges to the data distribution starting from the prior distribution. Beyond the generative modelling capacity of score-based models, we instead use the additionally encoded representations to perform inference tasks, such as classification.

In this work, we revisit the formulation provided by Abstreiter et al. (2021); Preechakul et al. (2022) which augments such diffusion-based systems with an encoder for performing representation learning which can be used for downstream tasks. In particular, we look at the infinite-dimensional representation learning methodology from Abstreiter et al. (2021) and perform a deeper dive into (a) the benefits of utilizing the trajectory or multiple points on it as opposed to choosing just a single point, and (b) the kind of information encoded at different points. Using trained attention

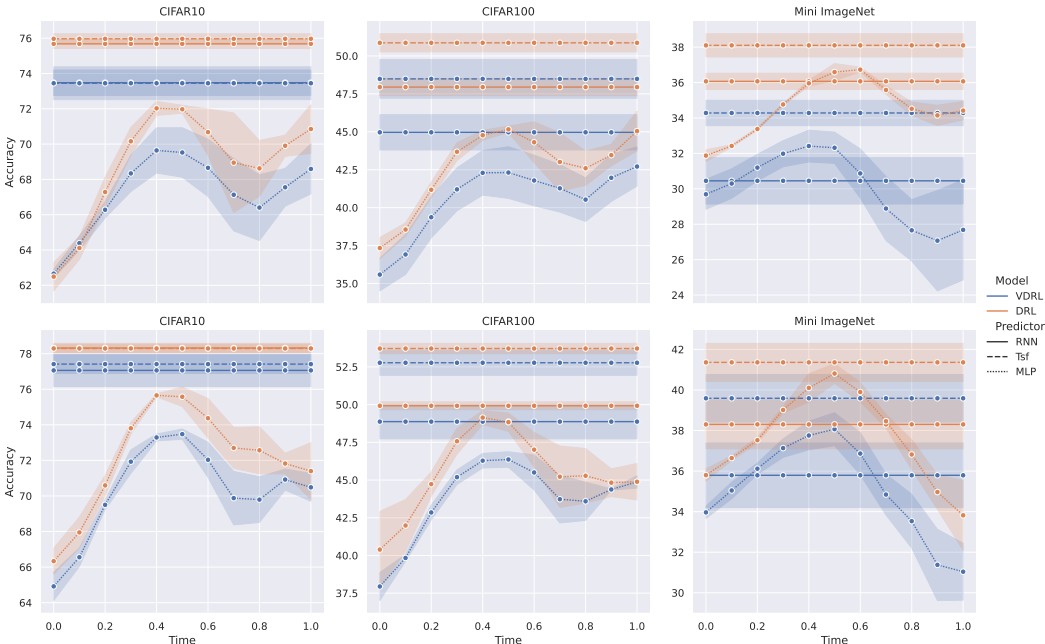

Figure 1: Downstream performance of single point based representations (MLP) and full trajectory based representations (RNN and Tsf) on different datasets for both types of learned encoders: probabilistic (VDRL) and deterministic (DRL) using a 64-dimensional latent space (*Top*) and a 128-dimensional latent space (*Bottom*).

mechanisms over diffusion trajectories, we ask about similarity and differences of representations across diffusion processes. Do they encode certain interpretable features at different points, or is it redundant to look at the whole trajectory? It is worth noting that while the representation itself is infinite-dimensional, we do discretize it for performing our analysis.

Our findings can be summarized as follows:

- We propose using the trajectory-based representation combined with sequential architectures like Recurrent Neural Networks (RNNs) and Transformers to perform downstream predictions using multiple points as it leads to better performance than just finding one-best point on the trajectory for downstream predictions (Abstreiter et al., 2021).

- We analyze the representations obtained at different parts of the trajectory through Mutual Information and Attention-based relevance to downstream tasks to showcase the differences in information contained along the trajectory.

- We also provide insights into the benefits of using more points on the trajectory, with saturating benefits as our discretization becomes finer. We further show that finer discretizations lead to even more performance benefits when the latent space is severely restricted, eg. just a 2-dimensional output from the encoder.

## 2 BEYOND FIXED REPRESENTATIONS

We first outline how diffusion-based representation learning systems are trained. Given some example $\mathbf{x}_0 \in \mathbb{R}^d$ which is sampled from the target distribution $p_0$, the diffusion process constructs the trajectory $(\mathbf{x}_t)_{t \in [0,1]}$ through the application of an SDE. In this work, we consider the Variance Exploding SDE (Song et al., 2021) for this diffusion process, defined as

$$d\mathbf{x} = f(\mathbf{x}, t) + g(t)d\mathbf{w} := \sqrt{\frac{d[\sigma^2(t)]}{dt}} d\mathbf{w} \qquad (1)$$

where $\mathbf{w}$ is the standard Wiener process and $\sigma^2(\cdot)$ the noise variance of the diffusion process. This leads to a closed form distribution of $\mathbf{x}_t$ conditional on $\mathbf{x}_0$ as $p_{0t}(\mathbf{x}_t|\mathbf{x}_0) = \mathcal{N}(\mathbf{x}_t; \mathbf{x}_0, [\sigma^2(t) -$

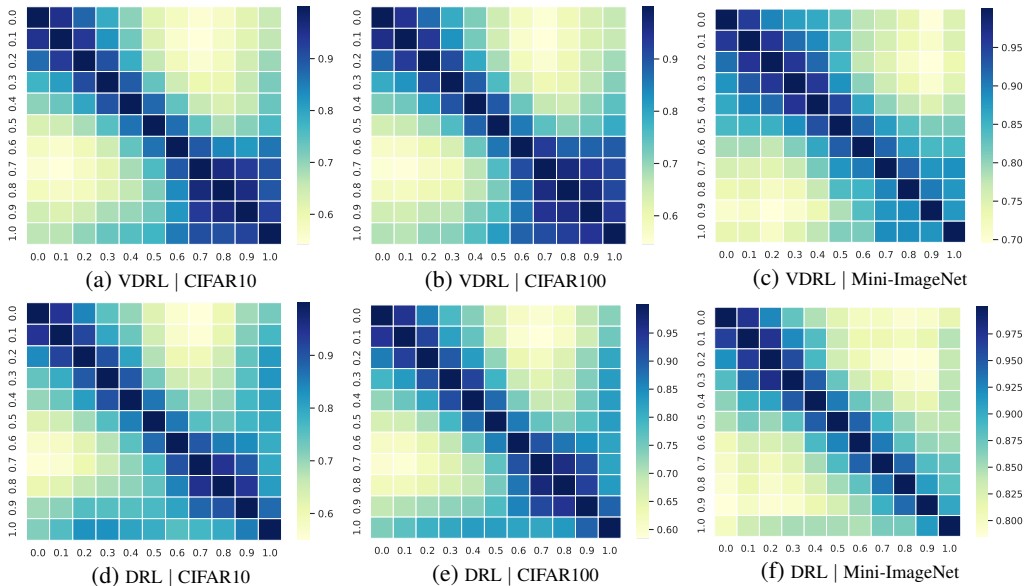

Figure 2: Normalized Mutual Information between different points on the trajectory. Cell $(i, j)$ demonstrates the normalized mutual information, estimated with the MINE algorithm, between the representations at time $t = i$ and $t = j$.

$\sigma^2(0)]\mathbf{I}$). Given this diffusion process modeled through the Variance Exploding SDE, the reverse SDE takes a similar form but requires the knowledge about the score function, i.e. $\nabla_{\mathbf{x}} \log p_t(\mathbf{x})$ for all $t \in [0, 1]$. A common way to obtain this score function is through the Explicit Score Matching (Hyvärinen & Dayan, 2005) objective,

$$\mathbb{E}_{\mathbf{x}_t} \left[ \|s_\theta(\mathbf{x}_t, t) - \nabla_{\mathbf{x}_t} \log p_t(\mathbf{x}_t)\|^2 \right] \tag{2}$$

which suffers from just one hiccup, which is that data about the ground-truth score function is not available. To solve this problem, Denoising Score Matching (Vincent, 2011) was proposed,

$$\mathbb{E}_{\mathbf{x}_0} \left[ \mathbb{E}_{\mathbf{x}_t|\mathbf{x}_0} \left[ \|s_\theta(\mathbf{x}_t, t) - \nabla_{\mathbf{x}_t} \log p_{0t}(\mathbf{x}_t|\mathbf{x}_0)\|^2 \right] \right] \tag{3}$$

where the term $\log p_{0t}(\mathbf{x}_t|\mathbf{x}_0)$ is available due to its closed-form structure. Given that the above objective cannot be reduced to 0, Abstreiter et al. (2021) proposes the objective

$$\mathbb{E}_{\mathbf{x}_0} \left[ \mathbb{E}_{\mathbf{x}_t|\mathbf{x}_0} \left[ \|s_\theta(\mathbf{x}_t, E_\phi(\mathbf{x}_0, t), t) - \nabla_{\mathbf{x}_t} \log p_{0t}(\mathbf{x}_t|\mathbf{x}_0)\|^2 \right] \right] \tag{4}$$

where the additional input $E_\phi(\mathbf{x}_0, t)$ to the score function is obtained from a learned encoder. It provides information about the unperturbed sample that might be useful for denoising data at time step $t$ in the diffusion process. Training this system can lead to the objective being reduced to 0, thereby providing incentive to the encoder $E_\phi(\cdot, t)$ to learn meaningful representations for each time $t$. From this, we obtain a trajectory-based representation $(E_\phi(\mathbf{x}_0, t))_{t \in [0,1]}$ for each sample $\mathbf{x}_0$, as opposed to finite sized representations obtained from typical Autoencoder (Bengio et al., 2013; Vinyals et al., 2016; Kingma & Welling, 2013; Rezende et al., 2014) and Contrastive Learning (Chen et al., 2020c; Grill et al., 2020; Caron et al., 2021; Bromley et al., 1993; Chen & He, 2020) approaches.

Following the setup in Abstreiter et al. (2021), we consider two different versions of the encoder $E_\phi(\cdot, \cdot)$, (a) the VDRL setup, where the output of $E_\phi(\cdot, \cdot)$ represents a distribution from which a sample is used, and the distribution is regularized using a KL-Divergence term with the standard Normal distribution $\mathcal{N}(0, \mathbf{I})$, and (b) the DRL setup, where the output of the encoder is deterministic and regularized using an $L_1$ distance metric to be as close to 0 as possible. Typically in all our experiments, we see that not only the trends hold with multiple seeds but also across these two types of encoders, substantiating the statistical significance of the trends.

It is important to note that our goal here is strictly representation learning, and thus we use the representations obtained for downstream (multitask-) image classification. This should not be confused with generative modelling as the provided mechanism augments a generative model for representation learning, but is not a generative model on its own. Since this representation learning

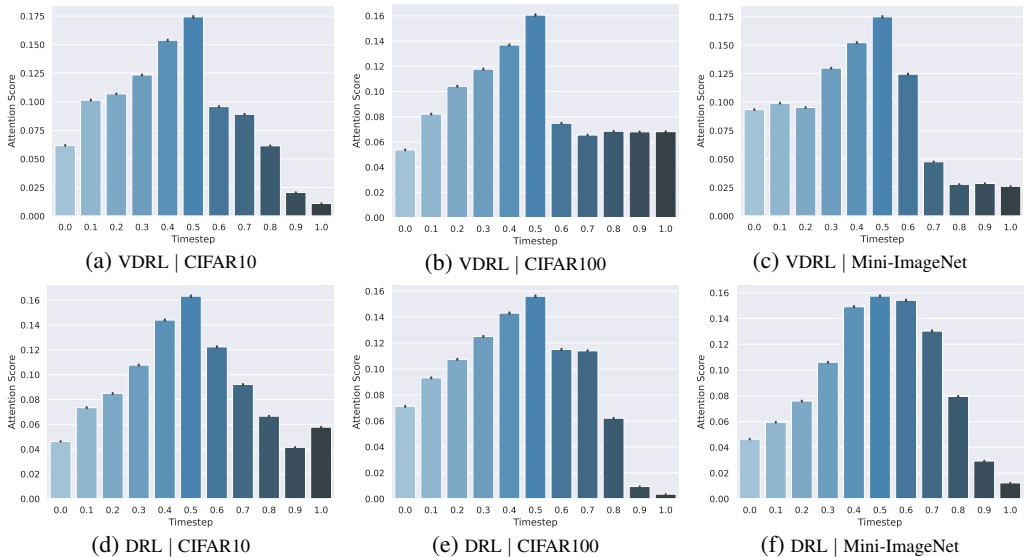

Figure 3: Attention Scores provided to different points on the trajectories, which are obtained from diffusion based representation learning systems with probabilistic encoders (VDRL; top row) and with deterministic encoders (DRL; bottom row) across the following datasets (i) Left: CIFAR10, (ii) Middle: CIFAR100, and (iii) Right: Mini-Imagenet.

paradigm can be augmented with a time-conditioned encoder model, this leads to a natural extension to trajectory-based (unbounded) representation, in contrast to typical bounded representation learning models like Autoencoders. Thus, this representation learning paradigm constructs a functional map from the input space to a curve / trajectory in $\mathbb{R}^d$, where we refer to $d$ as the dimensionality of this encoded space.

## 2.1 INFINITE-DIMENSIONAL REPRESENTATION OF FINITE-DIMENSIONAL DATA

Normally in autoencoders or other *static* representation learning methods, the input data $\mathbf{x}_0 \in \mathbb{R}^d$ is mapped to a single point $\mathbf{z} \in \mathbb{R}^c$ in the code space. However, our proposed algorithm learns a richer representation where the input $\mathbf{x}_0$ is mapped to a curve in $\mathbb{R}^c$ instead of a single point through the encoder $E_\phi(\cdot, t)$. Hence, the learned code is produced by the map $\mathbf{x}_0 \to (E_\phi(\mathbf{x}_0, t))_{t \in [0,1]}$ where the infinite-dimensional object $(E_\phi(\mathbf{x}_0, t))_{t \in [0,1]}$ is the encoding for $\mathbf{x}_0$.

The learned code is at least as good as static codes in terms of separation induced among the codes. Consider two input samples $\mathbf{x}_0$ and $\mathbf{x}_0'$, hence we have:

$$\|E_\phi(\mathbf{x}_0, 0) - E_\phi(\mathbf{x}_0', 0)\| \leq \sup_{t \in [0,1]} \|E_\phi(\mathbf{x}_0, t) - E_\phi(\mathbf{x}_0', t)\| \tag{5}$$

which implies that the downstream task can at least recover the separation provided by finite-dimensional codes from the infinite-dimensional code by looking for the maximum separation along the representation trajectory.

A downstream task can leverage this rich encoding in various ways. Consider the classification task where we want to find a mapping $f : \mathbb{R}^d \to \{0, 1\}$ from input data to the label space. Instead of giving $\mathbf{x}_0$ as the input to $f$, we define $f : \mathcal{H} \to \{0, 1\}$ where the input to the classifier is the whole trajectory $(E_\phi(\mathbf{x}_0, t))_{t \in [0,1]}$. Thus, the classifier can now use RNN and Transformer models to make use of the information content of the entire trajectories.

## 3 EXPERIMENTS

We first train two kinds of diffusion-based generative model as outlined in Abstreiter et al. (2021), based on probabilistic (VDRL) and deterministic (DRL) encoders respectively. After training, the encoder model is kept fixed. For all our downstream experiments, we use this trained encoder to obtain the trajectory based representation for each of the samples. While the trajectories lie in a continuous domain $[0, 1]$, we sample it at regular intervals with length 0.1, unless specified

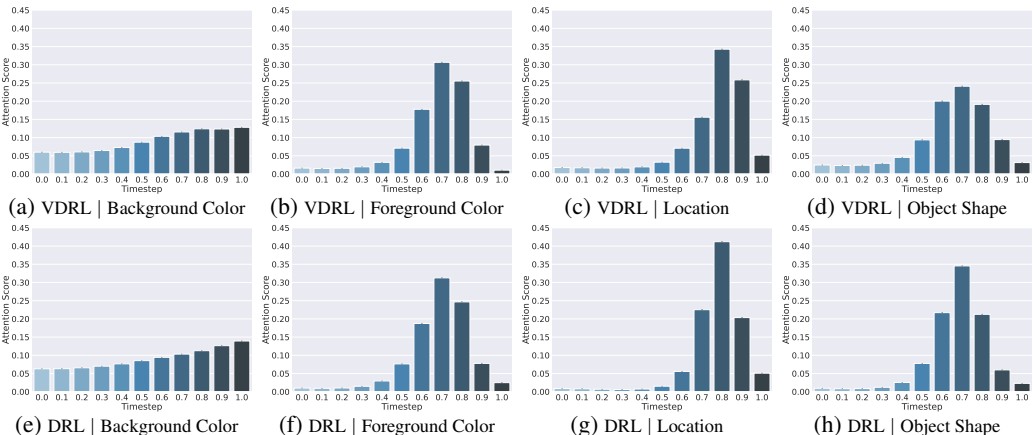

Figure 4: Attention score profiles for different tasks under the *Synthetic* dataset, when using the VDRL framework (top) and the DRL framework (bottom). The scores reveal that almost all points in the trajectory store similar amounts of information about the background color, while the latter part of the trajectory encodes more information about the foreground object. In particular, information about the location is most heavily found near the end of the trajectory.

otherwise. This leads to a discretization of the trajectory, which is then used for various analysis as outlined below. Further, we consider the dimensionality of the latent space, that is, the output of the encoder, as 128 unless otherwise specified. Additional details about the architectures used, the optimization strategy and other implementation details can be found in Appendix B.

## 3.1 DOWNSTREAM PERFORMANCE REVEALS BENEFITS OF TRAJECTORY INFORMATION

To understand the benefits of utilizing the trajectory-based representations, we train standard Multi-Layer Perceptron (MLP) models at different points on the trajectory and compare it with Recurrent Neural Network (RNN) (Hochreiter & Schmidhuber, 1997; Cho et al., 2014) and Transformer (Vaswani et al., 2017) based models that are able to aggregate information from different parts of the trajectory.

We evaluate the MLP, RNN and Transformer based downstream models on diffusion systems with both probabilistic encoders (VDRL) and also non-probabilistic ones (DRL). In Figure 1, we see the performance of these different setups for the following datasets: CIFAR10 (Krizhevsky et al., a), CIFAR100 (Krizhevsky et al., b) and Mini-ImageNet (Vinyals et al., 2016). Note that in contrast to MLP implementations, RNN and Transformer use the entire trajectory and the obtained performance is plotted across all time points for visual comparison. We typically see that RNN and Transformer based models perform better than even the peaks obtained by the MLP systems. This shows that there is no single point on the trajectory that encapsulates all the information necessary for optimal classification, and thus utilizing the whole trajectory as opposed to individual points leads to improvements in performance.

We further do this performance analysis for different dimensionality of the latent spaces, that is, when the trajectory representation is embedded in a 64-dimensional Euclidean space (Figure 1: Top) and when it is emebdded in a 128-dimensional Euclidean space (Figure 1: Bottom). We see similar trends across the two settings, thus highlighting consistent benefits when using a discretization of the whole trajectory.

## 3.2 MUTUAL INFORMATION REVEALS DIFFERENCES ALONG THE TRAJECTORY

In an effort to understand whether different parts of the trajectory based representation actually contain different types of information about the sample, we evaluate the mutual information between the representations at various points in the trajectory. We use the MINE algorithm (Belghazi et al., 2018) to estimate the mutual information between the representations at any two different points in the trajectory. Through this algorithm, we compute and analyse a normalized version of the mutual information, defined as $\mathrm{NMI}(\mathbf{X};\mathbf{Y}) := \mathrm{I}(\mathbf{X};\mathbf{Y})/\sqrt{\mathrm{H}(\mathbf{X})\mathrm{H}(\mathbf{Y})}$ where $\mathrm{I}(\cdot\,;\cdot)$ is the standard Mutual Information function (Cover, 1999) and $\mathrm{H}(\cdot)$ is the entropy function.

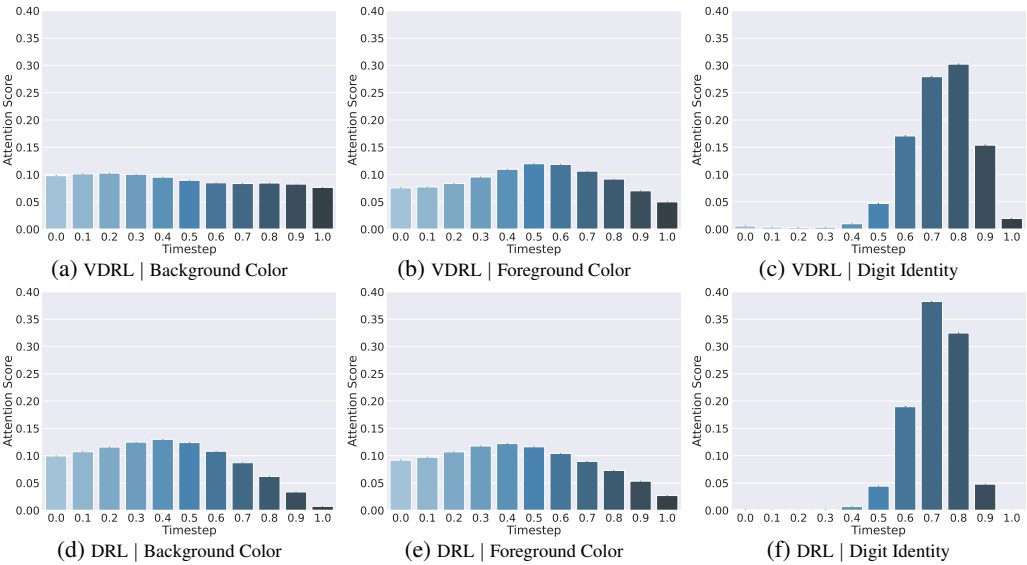

Figure 5: Attention score profiles for different tasks under the *Colored-MNIST* dataset, when using the VDRL framework (top) and the DRL framework (bottom). The scores reveal that almost all points in the trajectory store similar amounts of information about the background as well as the foreground color, while the latter part of the trajectory encodes more information about the identity of the digit.

Figure 2 illustrates the normalized mutual information between representations at different parts of the trajectory across three different datasets: CIFAR10, CIFAR100 and Mini-ImageNet as well as two different types of models: VDRL and DRL, where the former uses a probabilistic encoder and the latter doesn't. We see large normalized mutual information values near the principal diagonal and small values that are away from it, demonstrating that nearby representations on the trajectory are similar whereas distant points in the trajectory are considerably different. This shows that different parts of the trajectory learn to encode different kinds of information.

## 3.3 ATTENTION REVEALS RELEVANCE OF DIFFERENT PARTS OF THE TRAJECTORY

To complement the analysis in Sections 3.1 and 3.2, we train a single-layered Transformer model for downstream prediction, which comes from a learned embedding that queries information from different parts of the trajectory. Through the analysis of the attention scores at different points in the trajectory, we realize that the middle parts of the trajectory are the most important, as illustrated in the high attention scores around $t = 0.5$ in Figure 3.

This is in line with the performance results in Figure 1 which also shows that amongst the single-point MLP-based systems, the best downstream performance is reached near the middle of the trajectory. Attention score for any point in the trajectory, in a single-layered Transformer network, can be understood as the weight or importance of that point in the whole trajectory for the task in consideration. For the three image-classification datasets that we experiment on, we see that the attention patterns are quite similar. However, in later sections we will provide analysis with more controlled settings and see that the attention score profiles show varied behaviour for different features, indicating and strengthening the claim from Section 3.2 that the trajectory indeed encodes different information at different points.

## 3.4 PARSING SEMANTIC INFORMATION ENCODED ALONG THE TRAJECTORY

To better understand the different kind of information encoded in different parts of the trajectory, we expand our analysis into multi-task domains where each task relies on information from different features in the input. We consider three different datasets for this fine-grained analysis: *Synthetic*, *Colored-MNIST* and *CelebA*.

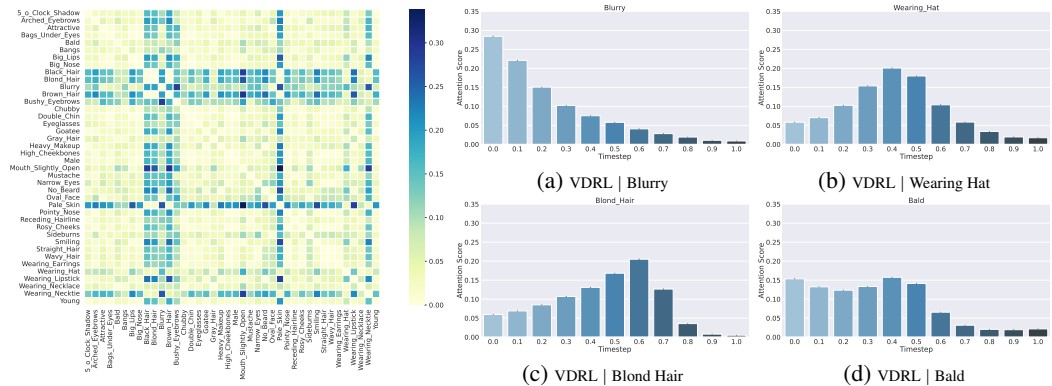

Figure 8: **Left**: Jensen Shannon Divergence plot for the attention profiles for any pair of features in the *CelebA* dataset; **Right**: Attention Profile plots for a subset of features in the *CelebA* dataset.

For all the analysis performed here, we train the diffusion model with the time-dependent encoder using Equation 4, and then keep it frozen. We then perform inference to obtain the trajectory representations for different granularities for each data point. Then, for each task in the dataset, we train a different single-layered transformer model and obtain attention scores over the trajectory corresponding to that particular task. This attention score over the points on the trajectory encodes the relevance of that area of the trajectory for the task, thereby providing insights on how much information about a particular feature is encoded in which part of the trajectory.

**Synthetic**. Synthetic dataset consists of an object in a scene. The scene consists of a distinct background color and the object is associated with a distinct foreground, location as well as the object shape. The system consists of four tasks; determining the (a) background color, (b) foreground color, (c) object location, and (d) object shape.

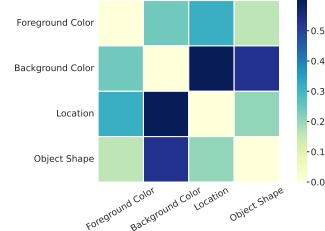

Figure 4 highlights the relevance of different parts of the trajectories for the different tasks. We see that while the background color information is more or less uniformly diffused over the whole trajectory, foreground information like object color, location and shape have a much more peaky distribution.

We also refer the readers to Figure 6 which highlights the differences in attention distributions over the trajectory. Cell $(i, j)$ in the figure refers to the Jensen-Shannon divergence (JSD) between the distribution over trajectory obtained for task $i$ with that obtained for task $j$.

Figure 6: Jensen Shannon Divergence plot for the attention profiles obtained for any pair of features for the *Synthetic* dataset using VDRL encoder.

For additional details about the analysis as well as additional ablations and results using different granularities and latent dimension sizes, please check out Appendix D. We also provide examples of samples from this setup in the Appendix.

**Colored-MNIST**. We then extend our analysis to a slightly more complex setting, where each sample consists of a digit with a distinct foreground color and digit identity (0-9), along with a background color on which the digit is embedded. The multi-task setting here consists of determining the (a) background color, (b) foreground color, and (c) digit identity.

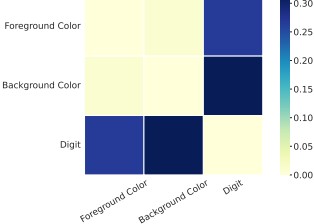

Figure 5 shows which parts of the trajectory are important for which task, and thus encode what kind of information. In this setting, we see that both the background color and the foreground color information is quite diffused over the trajectory, while the information about the digit identity is heavily present in the later parts of the trajectory.

Figure 7: Jensen Shannon Divergence plot for the attention profiles for any pair of features for the *Colored-MNIST* dataset using VDRL encoder.

We also refer the readers to Figure 7 for a birds-eye view on the differences in attention distributions over the trajectory for different tasks. Each cell $(i, j)$ in the figure refers to the JSD between the

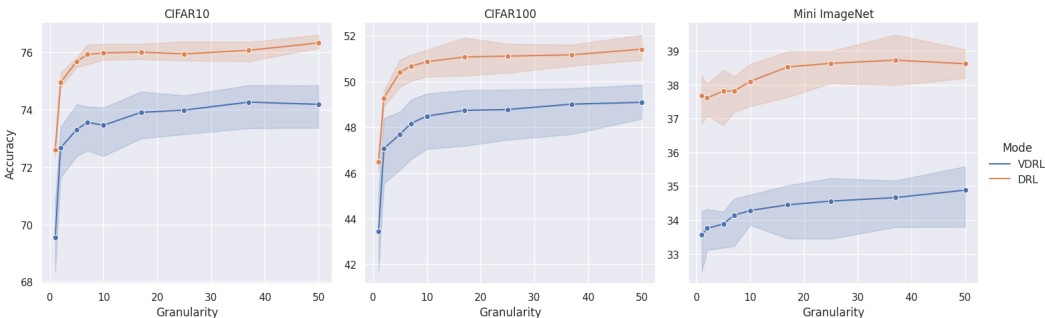

Figure 9: Performance on different datasets evaluated at different granularities of the trajectory representation. Granularity-$k$ on the X-axis implies that the discretization of the trajectory representation (obtained from the same frozen diffusion model), is done at $k$ uniform points, which are then fed to a Transformer model for downstream classification.

distribution over trajectory obtained for task $i$ with that obtained for task $j$. A high divergence implies that the information corresponding to the two tasks is not present together simultaneously.

For additional details about the analysis as well as additional ablations and results using different granularities and latent dimension sizes, please check out Appendix E. We also provide examples of samples from this setup in the Appendix.

**CelebA**. Finally, we conduct experiments on the CelebA dataset (Liu et al., 2015), which is a large-scale dataset that consists of images of celebrities as well as multiple binary labels for each corresponding to the different attributes; eg. whether the celebrity in the image has brown hair or not? We consider experimentation on CelebA to understand if we can semantically understand what kind of features, (in a more real world setting), are encoded in different parts of the trajectory. For a full list of the attributes in consideration in this dataset, we refer the readers to Appendix F.

We refer the readers to Figure 27 for a similar analysis of JSD between distributions over trajectories encoded by different features (or more formally; tasks corresponding to different features). While we do see some clustering (eg. blond hair, black hair and brown hair all have similar attention profiles), there is also a lot of uniformity in the divergences. We believe that scaling and extending this setting to richer and more diverse multi-task, multi-feature domains would allow for a much richer semantic separation between features.

We also highlight the individual attention profiles in Figure 27 for a subset of the features. It shows that the distributions learned for different features are actually different, implying presence of complementary information along the trajectory (Figures 29 and 30 in the Appendix).

Overall, we highlight how information about different features is encoded in different regions of the trajectory. This can be leveraged by learning an automated task-conditioned system that learns to "look in more detail" at certain parts of the trajectory while ignoring the others. For additional details about the analysis as well as additional ablations, please check out Appendix F.

## 3.5 BENEFITS OF USING MORE POINTS IN THE TRAJECTORY

We now extend our analysis to understand the benefits of having more points in the trajectory, that is, of moving closer to the continuous-time domain. We do this by considering more and more fine-grained discretizations of the trajectories, which we denote as granularity. A granularity of $k$ discretizes each trajectory representation by uniformly querying it at $(k+1)$ different points, in $[0, 1]$. Thus, a granularity level of 2 indicates using the points $\{0.0, 0.5, 1.0\}$ for downstream predictions.

We refer the readers to Figure 9 which illustrates the benefits of having more points in the trajectory. In particular, even though a granularity level of 2 has access to the mid-point of the trajectory, it doesn't do as well as when using a larger granularity. However, we do notice that the benefits to having more points in the trajectory do start to saturate beyond a certain point.

We believe that the dimensionality of the latent code might have a significant effect on the saturating point. To this end, we perform experimentation on low-dimensional trajectories on the synthetic datasets in the next section.

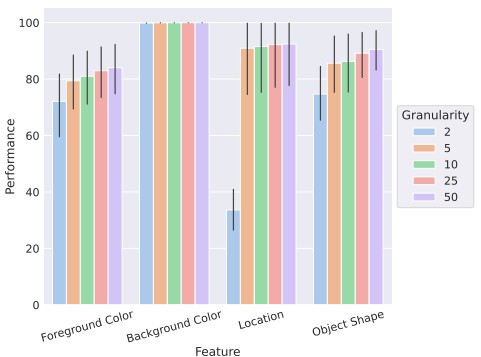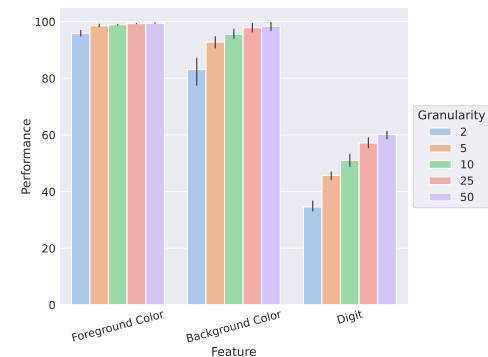

Figure 10: Improvements of increased granularity (finer discretization schedule) on different features for the Synthetic Dataset (*Left*) and the Colored-MNIST Dataset (*Right*). The dimensionality of the latent space is heavily restricted to 2, and hence we see that with the coarsest discretization, performance on a number of features suffers.

## 3.6 INTERPLAY BETWEEN CODE-DIMENSIONALITY AND GRANULARITY

To provide additional analysis into the benefits of having this infinite-dimensional representation, we consider the *Synthetic* and *Colored-MNIST* datasets and use a very small dimensionality for the latent space, which is the output of the encoder. In particular, we consider the output of the encoder to be a 2-dimensional code, but unbounded in the time domain.

We discretize this 2-dimensional trajectory representation at different uniform points, similar to the analysis done in Section 3.5 to see how well a limited capacity (in latent bits per point on trajectory) code can do and how much does it benefit from the increase in dimensions through the time dimension?

Figure 10 highlights that for both the datasets, we see substantial improvement when using more points on the trajectory when the latent code is severely restricted, further signifying the benefit of the temporal unbounded-ness of the trajectory, and the monotonic improvement in performance with increase in granularity.

## 4 CONCLUSION

Through our analysis, we realize that the encoder $E_\phi(\cdot, t)$ actually learns different kinds of information at different time-steps $t$. Typically the mid-points of the trajectory are the most important for downstream classification tasks but we uncover that using as many points on the whole trajectory, i.e. increasingly finer discretization of an infinite-dimensional object, is much better than just singular points on it. What kind of semantic information is encoded in the different parts of the trajectories? Can we show some benefits of the unboundedness of the trajectory? Through our analysis, we provide insights into the differences of information stored along the trajectory, as well as the benefits of its unbouneded structure especially in the domain of restricted latent dimensionality.

While we highlight some interesting properties of these trajectory-based representations as well as the diversity of information over it, we believe that an important next step is to automate and learn the discretization process as opposed to the heuristic based uniform discretization. We believe this could lead to a variable computation system, where the downstream model would learn on its own which part of the trajectory should it sample more finely than others, when conditioned on the task/feature used.

This kind of task-conditioned discretization process would not only be able to use the whole trajectory information without heuristics but would also be able to leverage the structure which we show in our analysis in a more efficient and improved manner. We believe that this is an important direction to obtaining task and feature centric representations which are more general than the finite-sized representations afforded by contemporary representation learning models.

## ETHICS STATEMENT

We do not foresee any negative or unethical implications of this work, which is in addition to the general impacts of advancement of Machine Learning and Representation Learning.

## REPRODUCIBILITY STATEMENT

We perform all the experiments with multiple seeds, ranging from 3 to 10 depending on the experiments. For each run of the score-based diffusion model, we also perform the downstream experiments with multiple seeds to obtain statistically significant results. We refer the readers to the implementation details outlined in Appendix B and we will be open-sourcing our code for ease of reproducibility.

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

APPENDIX

## A    CONNECTION TO SDEDIT

Our work has a close and reciprocal connection to Meng et al. (2021) that we refer to as SDEdit for convenience. SDEdit has been proposed as an image editing algorithm that transforms a *guide* image $x^{(g)}$ which can contain different levels of details (e.g. coarse colored strokes or colored strokes on real images) to a realistic image that is to some extent faithful to the guide. In short, the way SDEdit performs this task is as the following:

1. Train a normal diffusion model on the original image dataset. Let $s_\theta(x, t)$ be the learned time-dependent score function.

2. Add noise to the guide image so that it looks like a sample from an intermediate distribution of the diffusion process at time $t_0 \in (0, 1)$:

$$x^{(g)} \sim (N)(x^{(g)}, \sigma^2(t_0)I).$$

3. Then start the reverse diffusion process from the noisy guide at time $t_0$ (instead of starting from pure noise) to obtain a realistic image which is conditioned on the guide via the starting point. Let

$$\text{SDEdit}(x^{(g)}; t_0, \theta)$$

be the output of this process.

A critical hyper-parameter of this algorithm is $t_0$ which is chosen empirically in Meng et al. (2021) from within [0.3, 0.6]. We show in the following a bidirectional connection between SDEdit and the infinite-dimensional trajectory-based representation of our work.

**From SDEdit → Ours:**   Notice that the guide $x^{(g)}$ can be an abstract semantic description of the image, e.g. the left-most column of Figure 4 in Meng et al. (2021) may be viewed as a semantic segmentation of the image which is intended to be generated. The optimal choice for $t_0$ corresponds to the amount of noise added to the clean guide such that $x^{(g)}$ is still faithful to the original guide but at the same time is similar to the original images contaminated with the same amount of noise. Let $P_0$ be the distribution of the original images and $x(0) \sim P_0$ be a clean image. The time $t_0$ is chosen such that $x^{(g)}(t_0) \sim P_{t_0}$. In other words, both original image $x(t_0)$ and the guide $x^{(g)}(t_0)$ contains similar information content at time $t_0$. This means that by starting the forward diffusion process from the clean image $x$, the information content of the guide is most conspicuous in $x(t_0)$. Put it differently, consider the reverse procedure of SDEdit which instead of making a realistic image from the guide, the goal is to estimate the guide from the realistic image (e.g. doing semantic segmentation from a clean image.) Assume the estimation (segmentation) model is free to run a diffusion process on the clean image $x$ and choose an intermediate time within (0, 1) which is best to predict the guide (semantic segmentation). The above reasoning implies that the model chooses $t_0$.

**From Ours → SDEdit:**   The core idea of the present work was to investigate the information content of the representations corresponding to different time steps along the trajectories of the diffusion process. This was done by defining a downstream task whose solution requires extracting some abstract information from the images (e.g. location, identity or colour of the objects in the image) allowing the prediction model for this task to look at the entire trajectory as the information source instead of choosing the most useful intermediate time step. Our empirical results show that when the model is free to mix information from more than one time step, the mixing weight is not concentrated on a single time step and distributed smoothly over various time steps. The implication of this result for SDEdit is that instead of using $x^{(g)}(t_0)$ as the starting point of the reverse diffusion process, a mixing of $x^{(g)}(t_j), j \in J$ should be a more suitable starting point where $J$ is the a subset of time indices to which our downstream estimation model has attributed large weights. In this case, *more suitable* means that it entails a better trade-off between *faithfulness to the guide* and *realisticness* of the final image as defined in Meng et al. (2021).

## B    IMPLEMENTATION DETAILS

**Score Model:** We use the implementation of the score model from Song et al. (2021) using the variance-exploding SDE and the NCSN++ configuration, in particular a smaller version of the CIFAR10 configuration provided on their codebase. We augment the score-model with an Encoder $E_\phi$ which is implemented as the $28 \times 2$ Wide-ResNet architecture (Zagoruyko & Komodakis, 2016) that maps the input with time embeddings to a vector in $\mathbb{R}^d$, where $d$ is the dimensionality of the latent space and is set to 128 unless otherwise specified. The time embeddings for the encoder model are implemented in the exact same way as for the score inputs, as outlined in Song et al. (2021). We use the learning rate of $2 \times 10^{-4}$ to optimize the score network.

**Downstream Model:** For Multi-Layer-Perceptron (MLP) based Classification model, we consider a network with a single hidden layer, ReLU activation function, and 512 neurons. For the Recurrent Neural Network (RNN) model, we use a GRU with 256 hidden units and for the transformer system, we use a Multi-Head attention system with 4 heads and two layers, with weight sharing between the layers. For our attention profile based analysis settings, we consider the same Multi-Head attention system but only use a single layer instead of two, as it allows to make the score (averaged over heads) more interpretable.

We train all the downstream models with dropout of 0.25 and perform hyperparameter optimization for the learning rate over the set $\{0.001, 0.00075, 0.0005, 0.00025, 0.0001, 0.00005\}$. In particular, we found the hyperparameter optimization important when considering the granularity analysis.

## C    CIFAR10, CIFAR100 AND MINI-IMAGENET

We train the score model for 70,000 iterations and then the downstream models for 100 epochs. For the performance of the models, we use a $2-$layered Transformer model while for attention score profiles, we use a single layered Transformer model.

## D    SYNTHETIC

We train the score model for 250,000 iterations and then the downstream models for 1500 epochs. Figure 11 shows some samples obtained from this dataset, showcasing the different features present as well as the diversity of these different features.

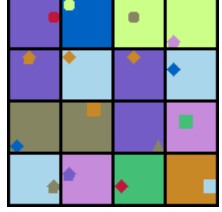

We additionally perform the Jensen-Shannon Divergence analysis between different features for different granularities, as well as visualize the attention score profiles for the different granularities as well. Furthermore, we do the same analysis with both the types of encoders; VDRL and DRL.

Figure 11: Samples from Synthetic Dataset

The corresponding plots for the attention score profiles are present in Figures 12 - 17 for different latent space dimensionalities, different granularities and the different types of encoding schemes (VDRL and DRL). Further analysis into the performance on different features with different granularities and dimensionalities can be found in Figure 18.

## E    COLORED MNIST

We train the score model for 250,000 iterations and then the downstream models for 1500 epochs. Figure 26 shows some samples obtained from this dataset, showcasing the different features present as well as the diversity of these different features.

We additionally perform the Jensen-Shannon Divergence analysis between different features for different granularities, as well as visualize the attention score profiles for the different granularities as well. Furthermore, we do the same analysis with both the types of encoders; VDRL and DRL.

Figure 26: Samples from Colored-MNIST Dataset

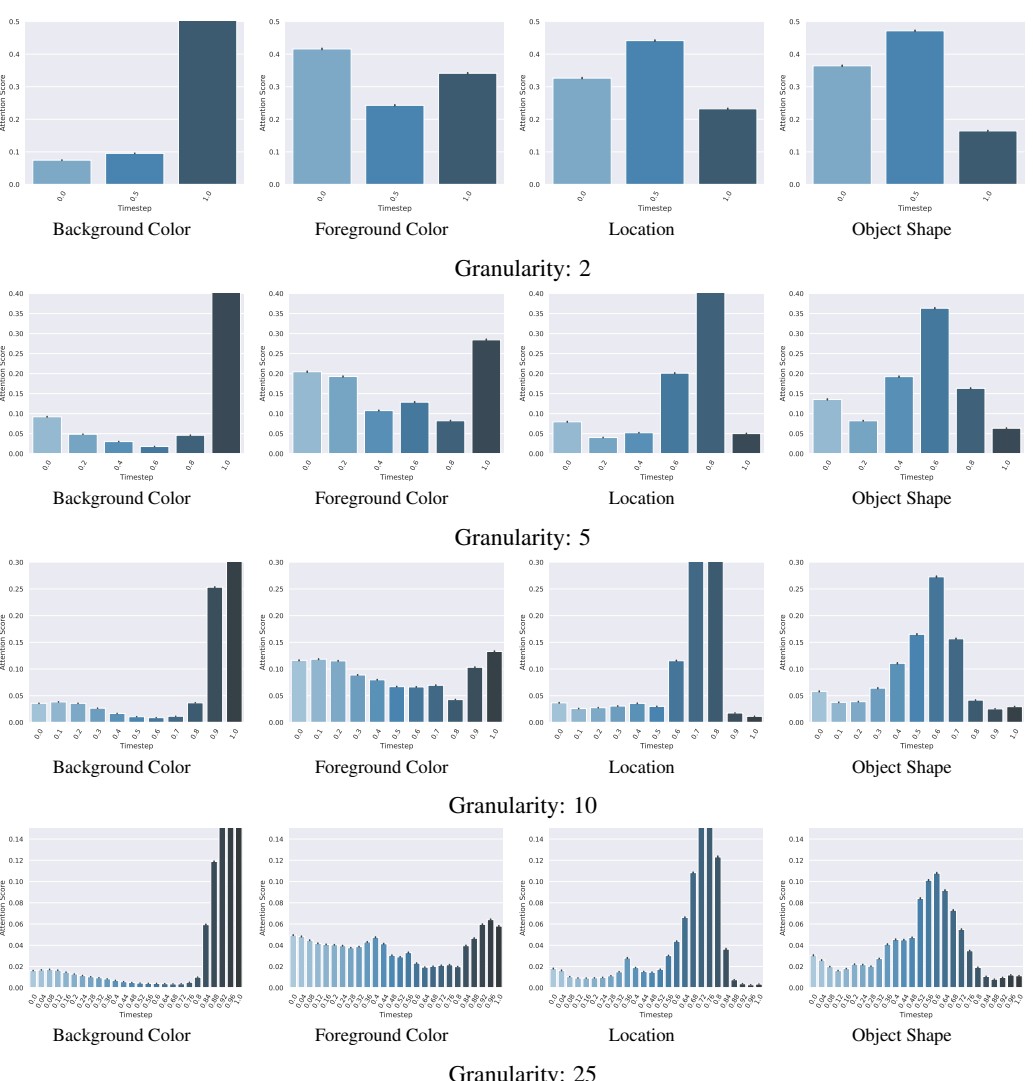

Figure 12: Attention score profiles for the synthetic dataset on the different features, using different granularities, with the dimensionality of the latent space as 2 and the VDRL encoder.

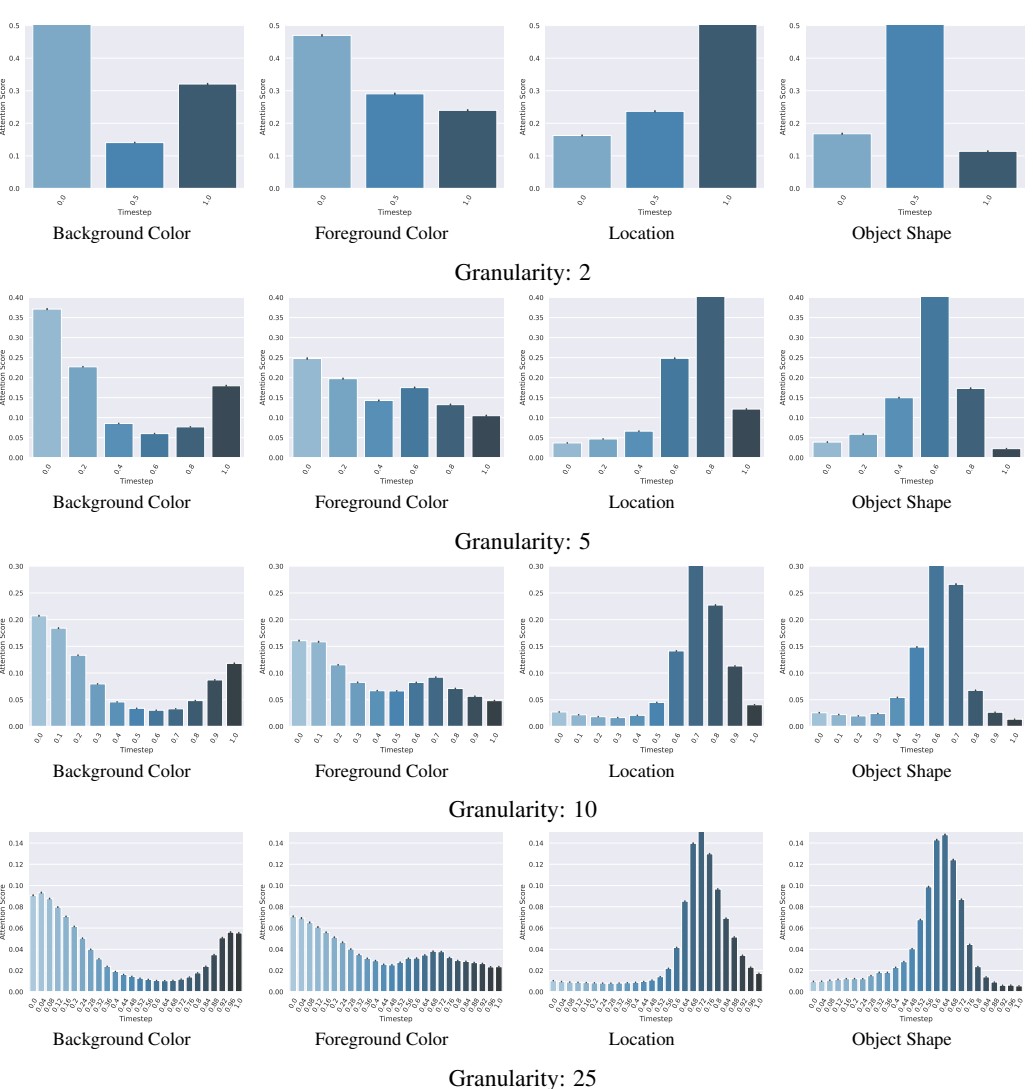

Figure 13: Attention score profiles for the synthetic dataset on the different features, using different granularities, with the dimensionality of the latent space as 16 and the VDRL encoder.

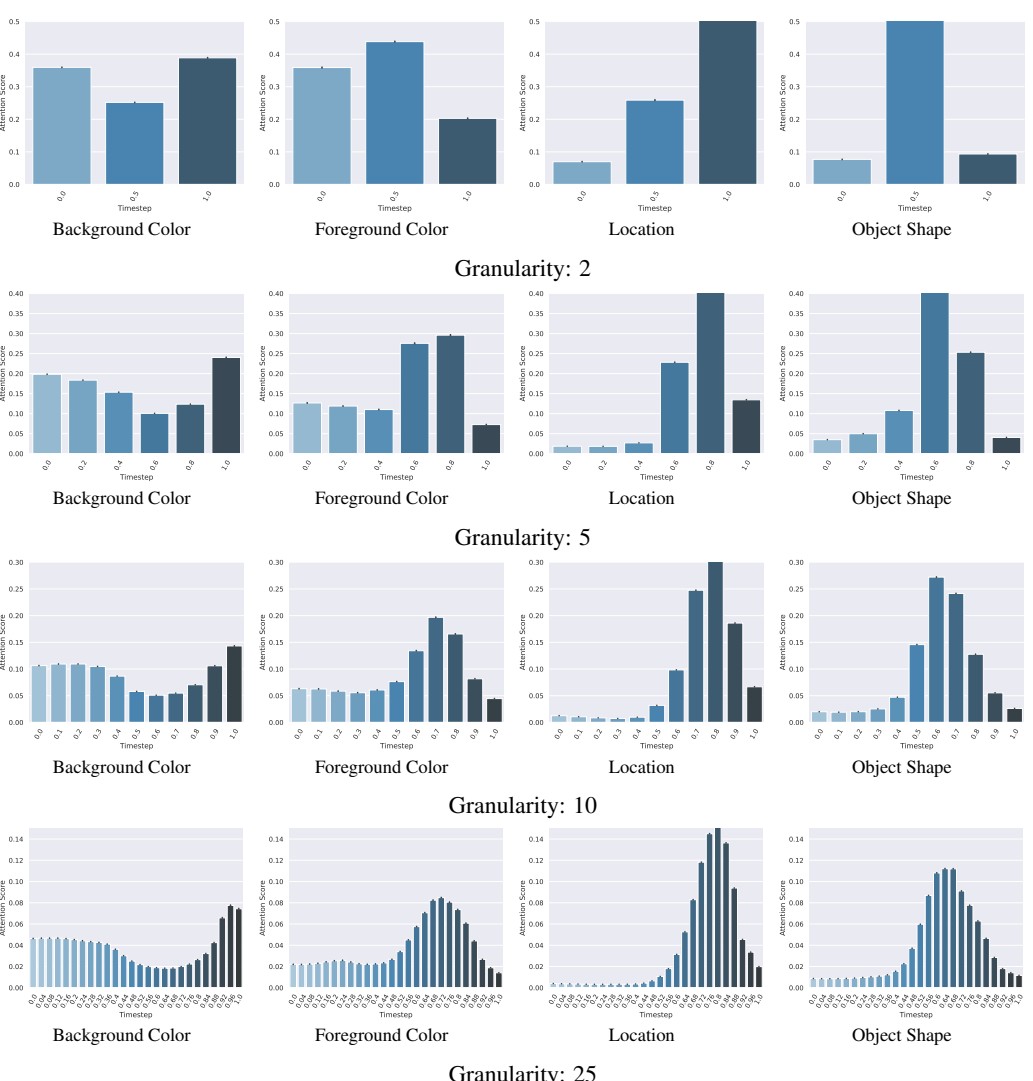

Figure 14: Attention score profiles for the synthetic dataset on the different features, using different granularities, with the dimensionality of the latent space as 32 and the VDRL encoder.

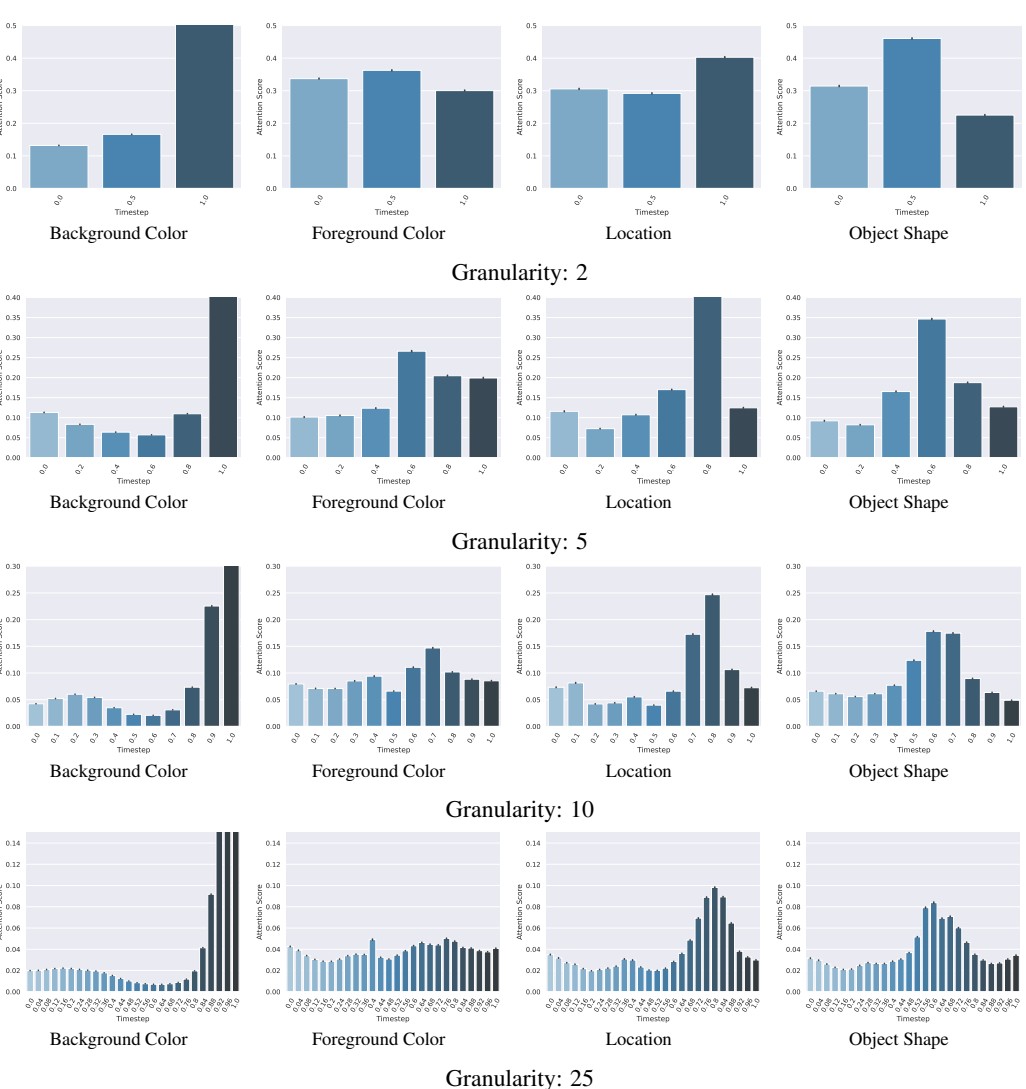

Figure 15: Attention score profiles for the synthetic dataset on the different features, using different granularities, with the dimensionality of the latent space as 2 and the DRL encoder.

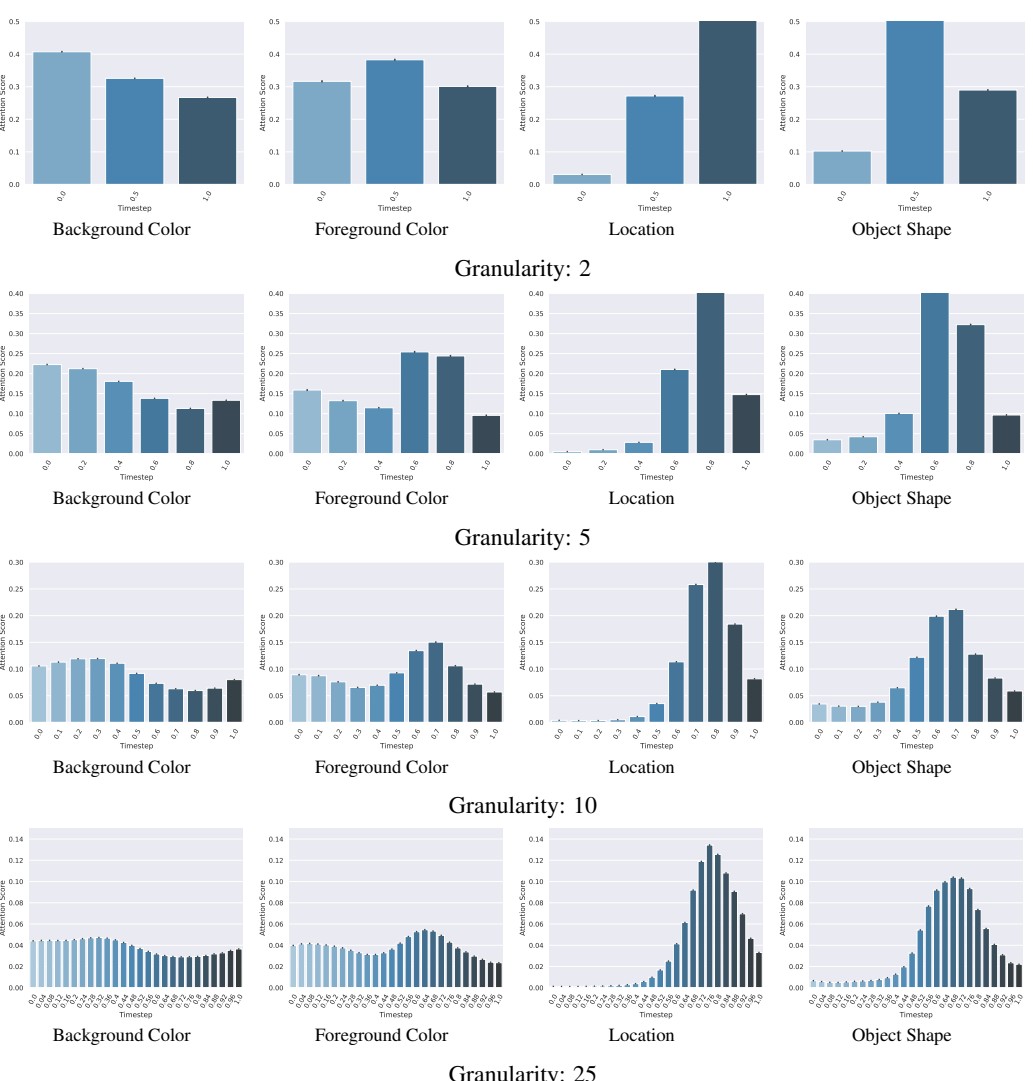

Figure 16: Attention score profiles for the synthetic dataset on the different features, using different granularities, with the dimensionality of the latent space as 16 and the DRL encoder.

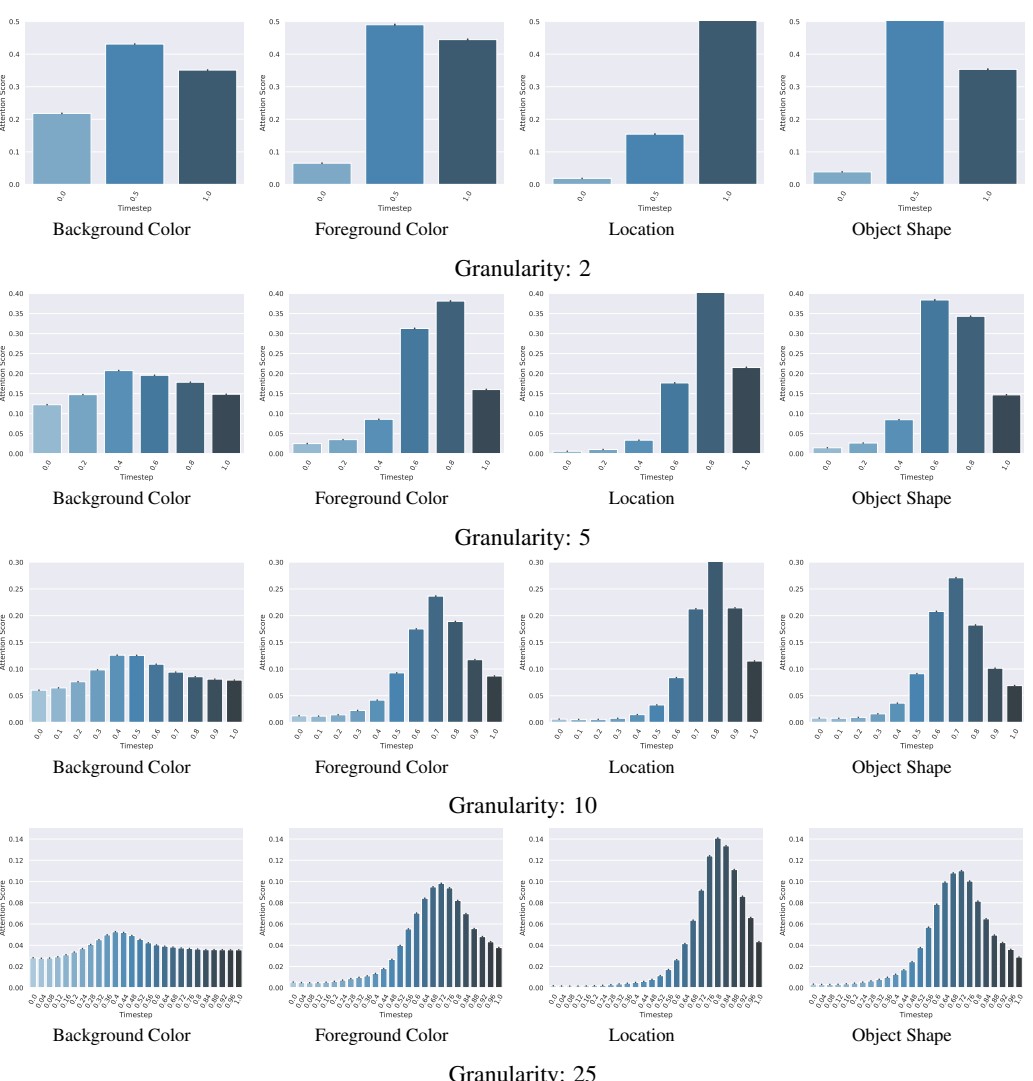

Figure 17: Attention score profiles for the synthetic dataset on the different features, using different granularities, with the dimensionality of the latent space as 32 and the DRL encoder.

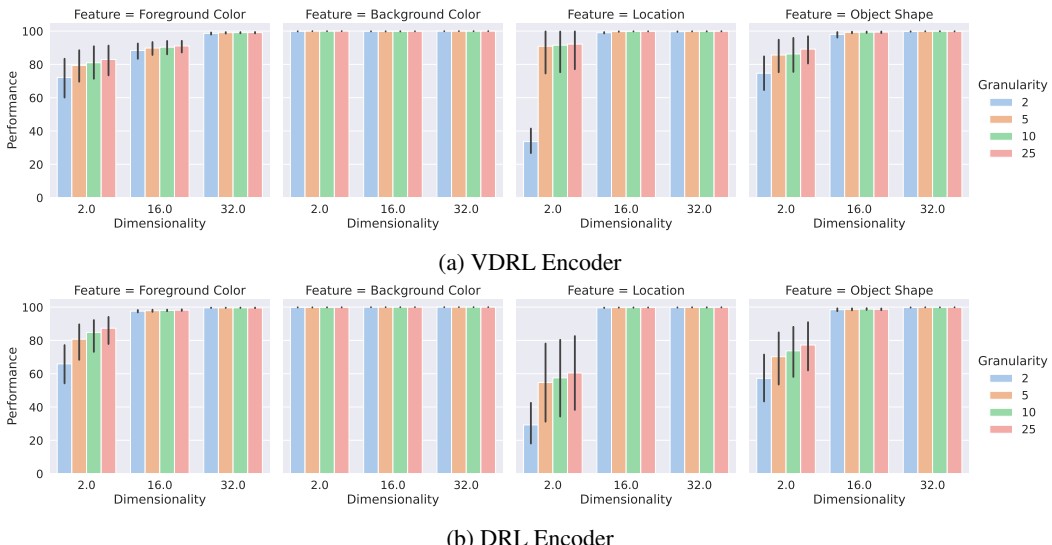

(a) VDRL Encoder

(b) DRL Encoder

Figure 18: Downstream performance plots for the Synthetic Dataset for different features, when the score model is trained with different latent dimensionality and the downstream models are trained with different granularities for discretization.

The corresponding plots for the attention score profiles are present in Figures 19 - 24 for different latent space dimensionalities, different granularities and the different types of encoding schemes (VDRL and DRL). Further analysis into the performance on different features with different granularities and dimensionalities can be found in Figure 25.

## F CELEBA

We train the score model for 250,000 iterations and then the downstream models for 100 epochs. We additionally perform the Jensen-Shannon Divergence analysis between different features for two different types of encoders; VDRL and DRL. The corresponding plots for these analysis, as well as for the attention score profiles and performances on different features, are present in Figures 27 - 30. The figures also enumerate the different attributes present in the dataset.

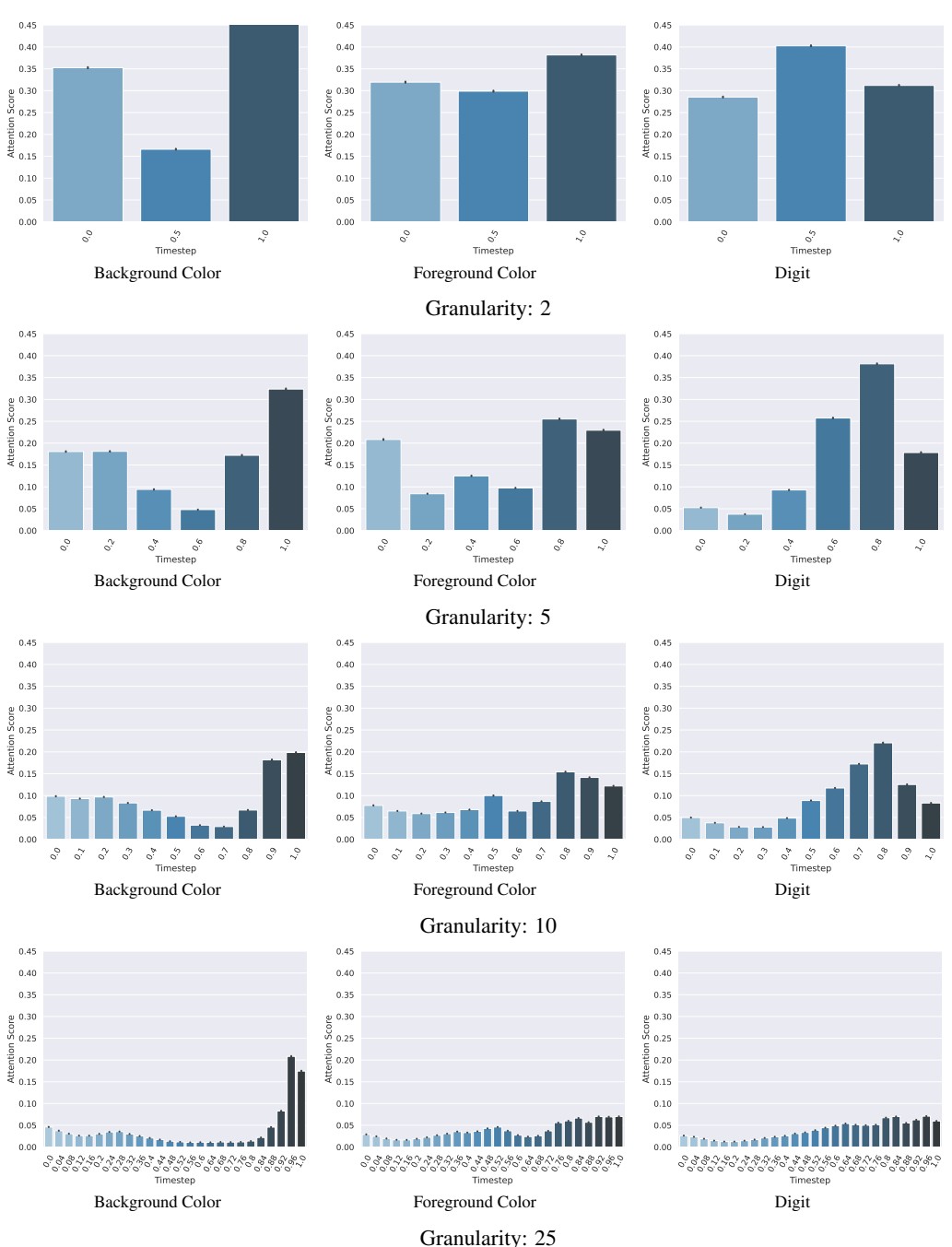

Figure 19: Attention score profiles for the Colored-MNIST dataset on the different features, using different granularities, with the dimensionality of the latent space as 2 and the VDRL encoder.

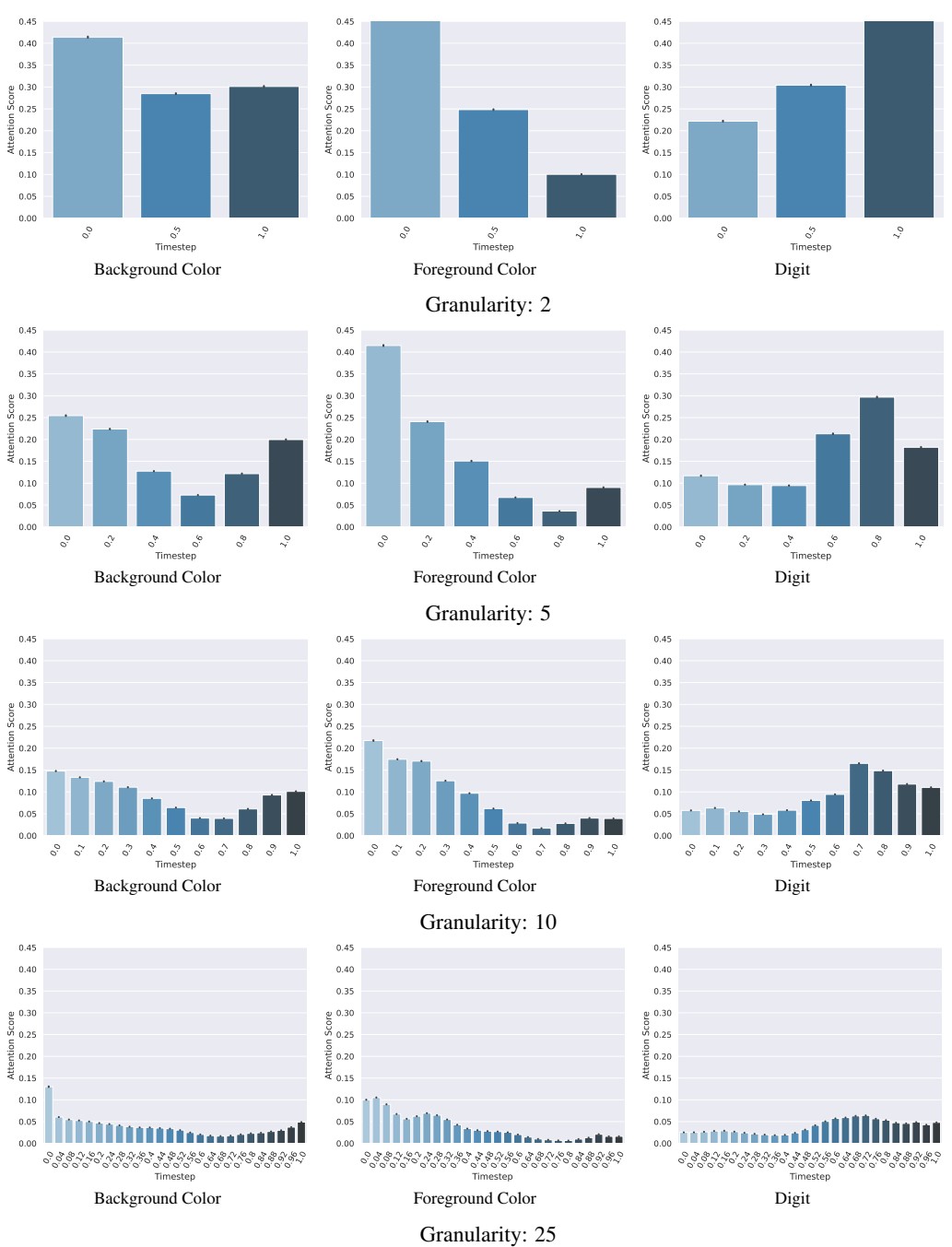

Figure 20: Attention score profiles for the Colored-MNIST dataset on the different features, using different granularities, with the dimensionality of the latent space as 16 and the VDRL encoder.

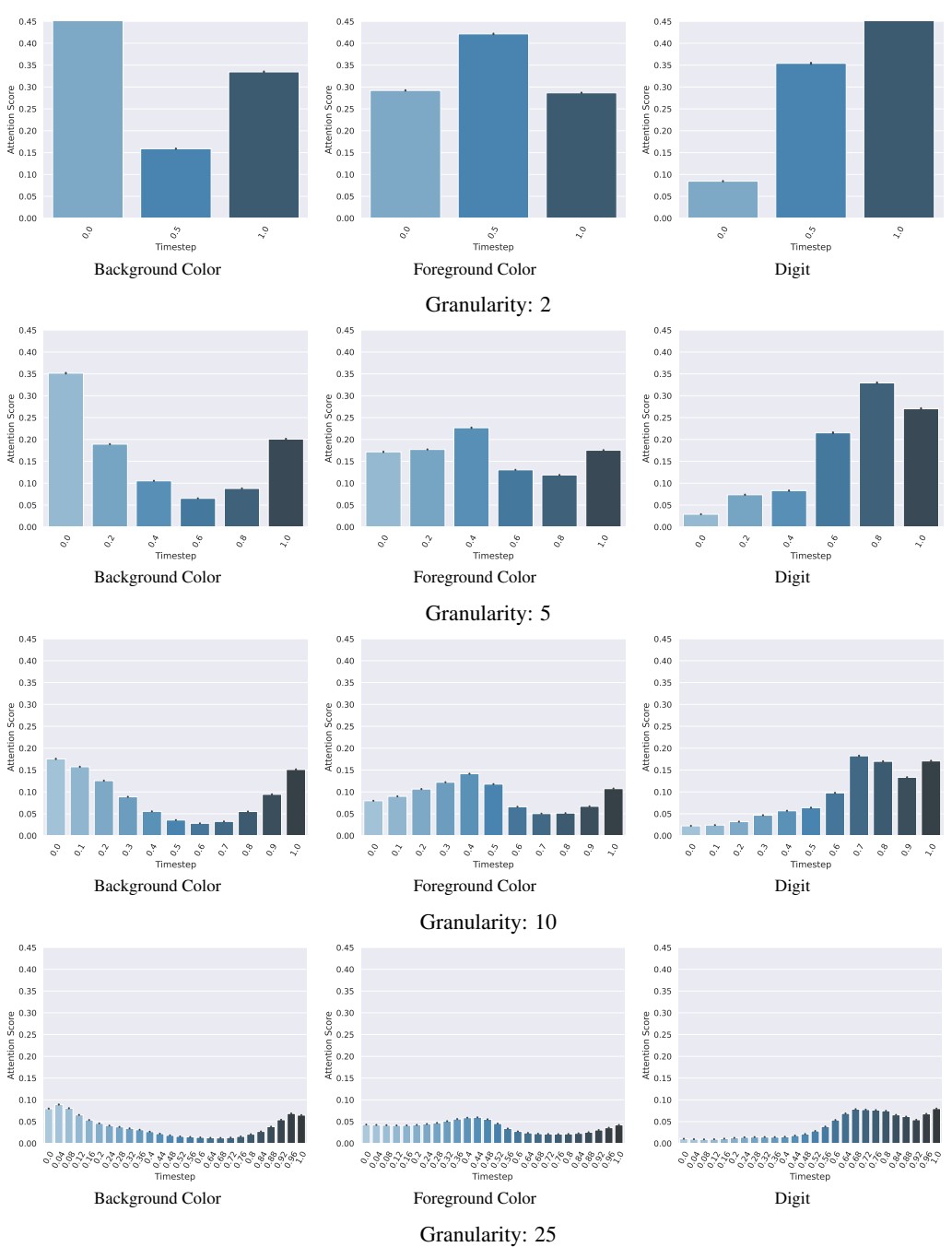

Figure 21: Attention score profiles for the Colored-MNIST dataset on the different features, using different granularities, with the dimensionality of the latent space as 32 and the VDRL encoder.

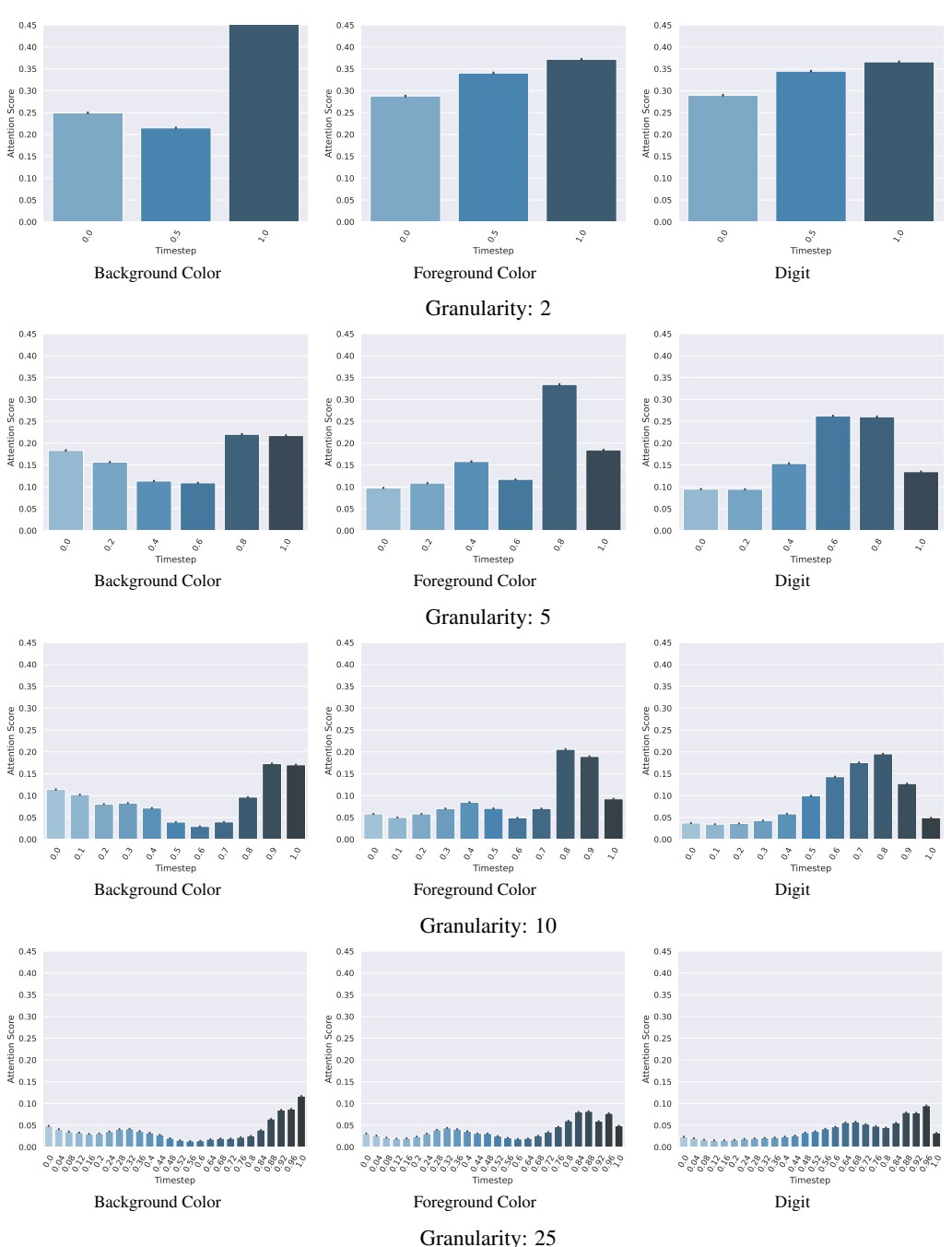

Figure 22: Attention score profiles for the Colored-MNIST dataset on the different features, using different granularities, with the dimensionality of the latent space as 2 and the DRL encoder.

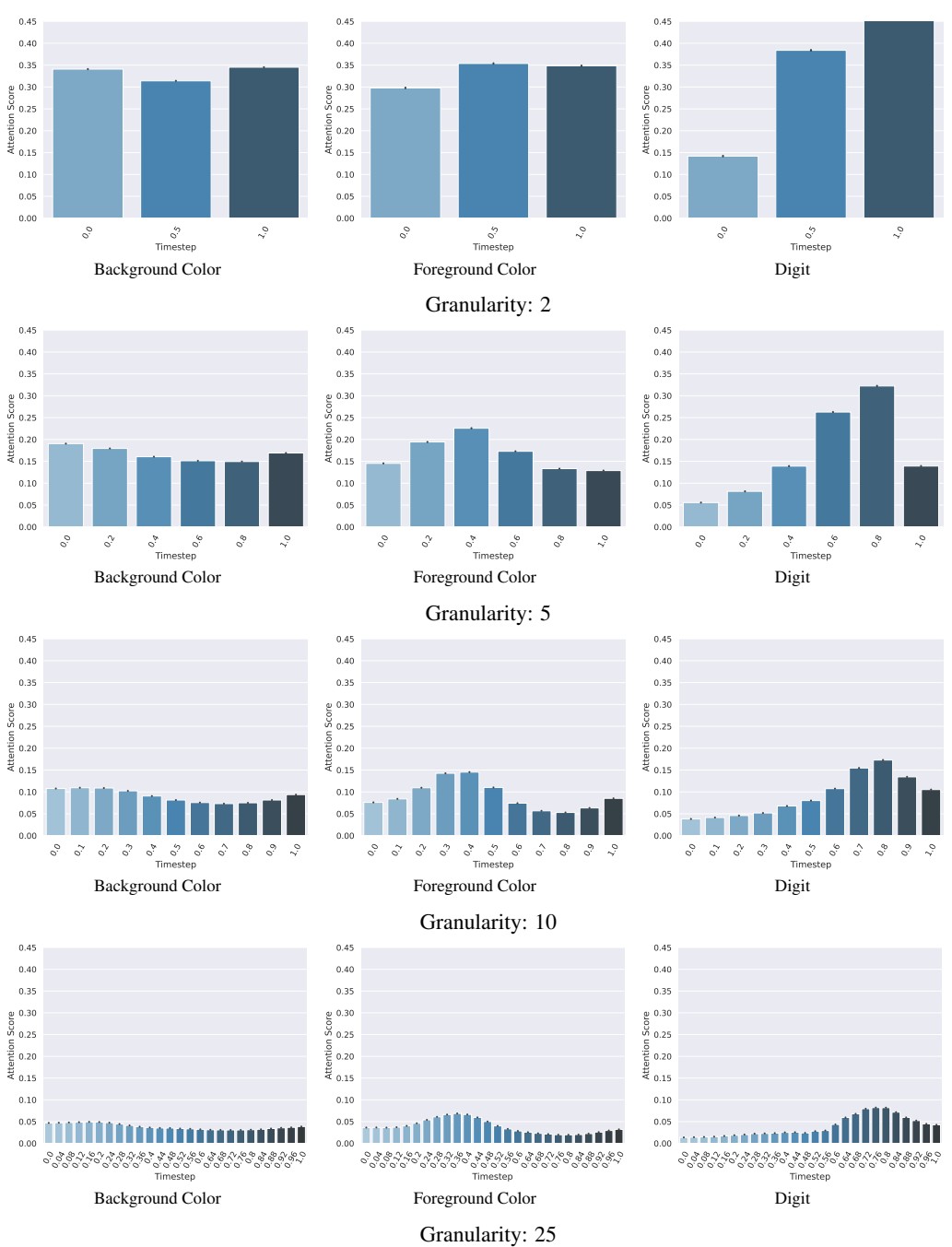

Figure 23: Attention score profiles for the Colored-MNIST dataset on the different features, using different granularities, with the dimensionality of the latent space as 16 and the DRL encoder.

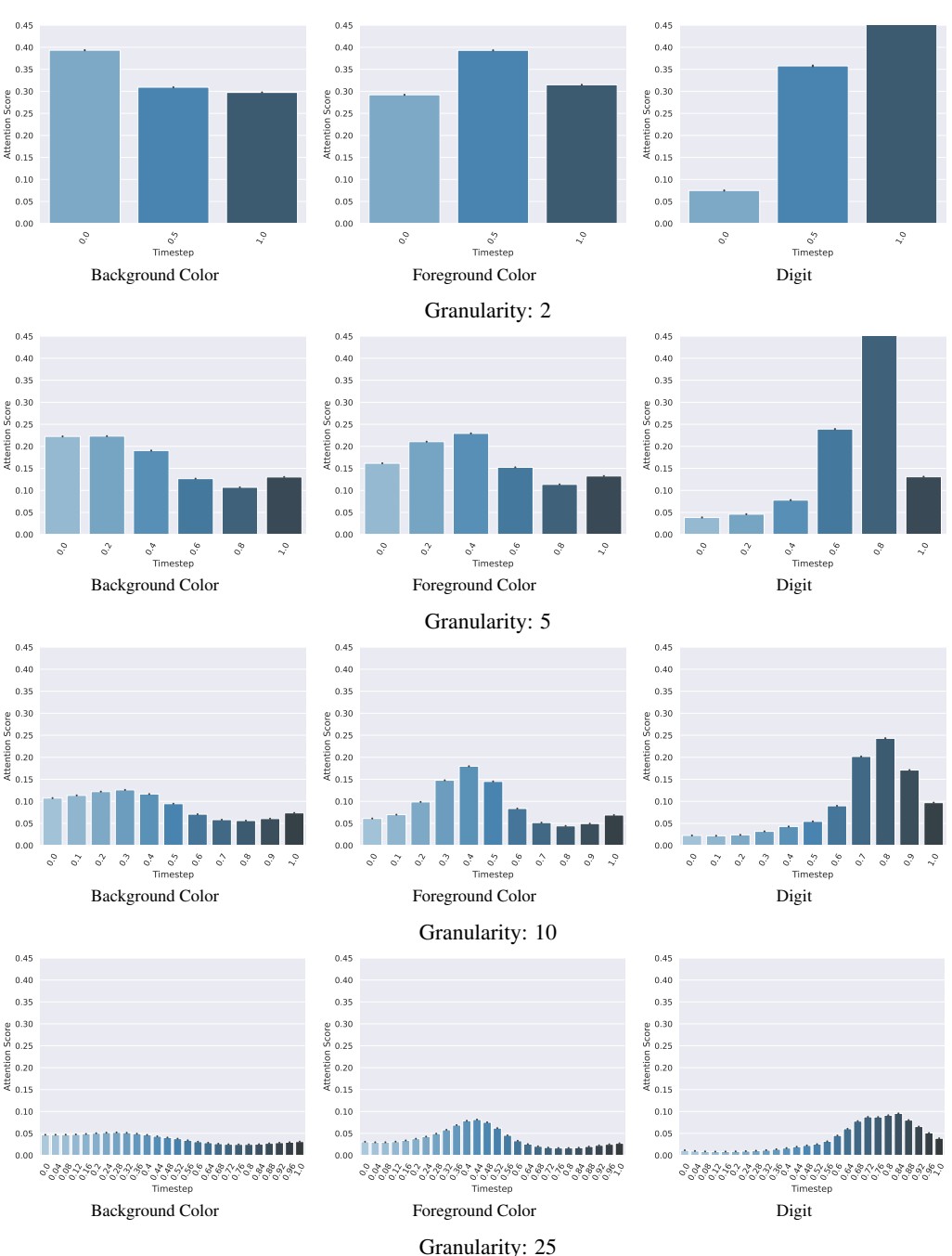

Figure 24: Attention score profiles for the Colored-MNIST dataset on the different features, using different granularities, with the dimensionality of the latent space as 32 and the DRL encoder.

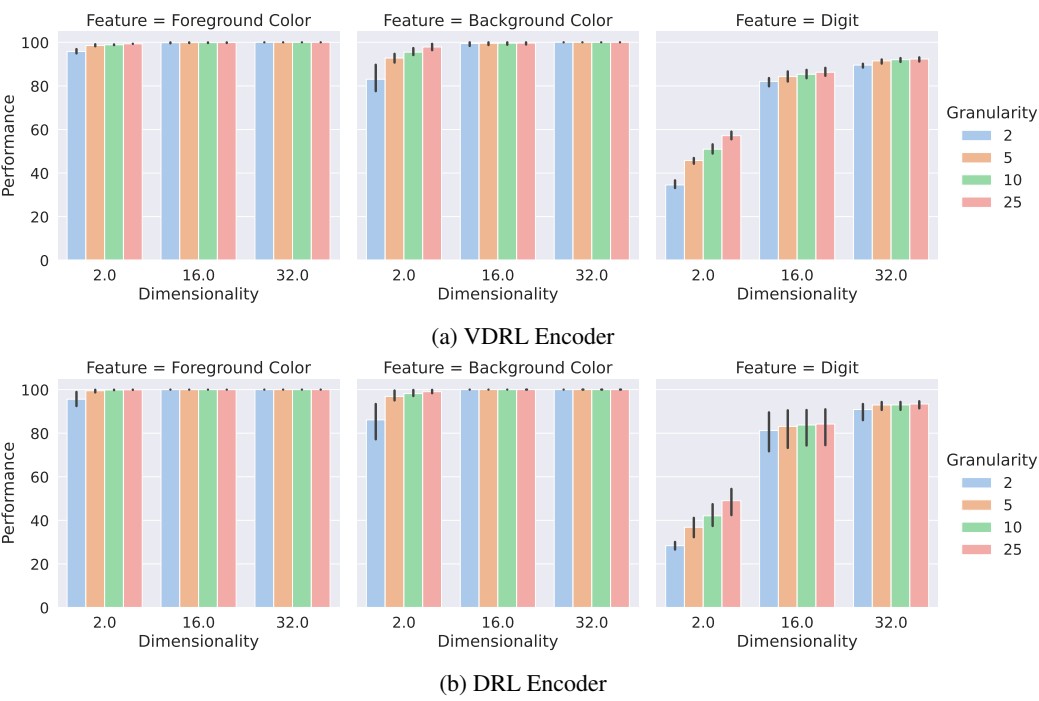

Figure 25: Downstream performance plots for the Colored-MNIST Dataset for different features, when the score model is trained with different latent dimensionality and the downstream models are trained with different granularities for discretization.

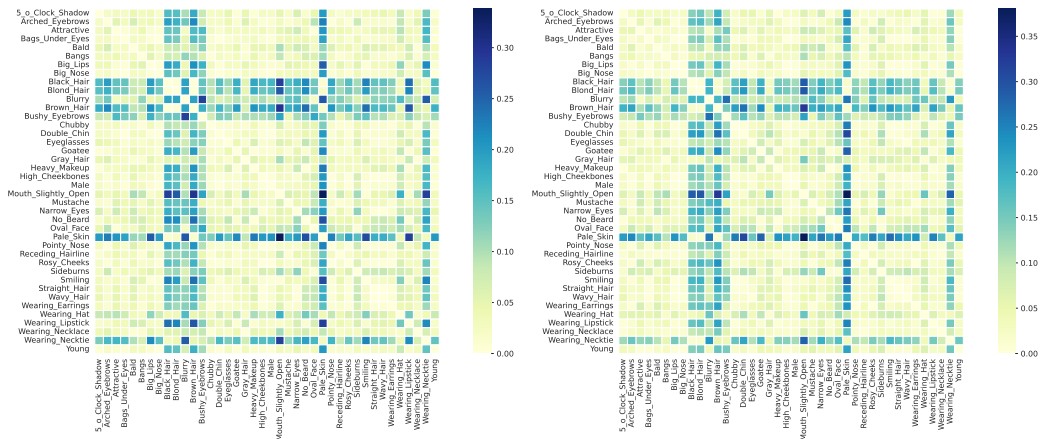

Figure 27: Jensen Shannon Divergence plot for the attention profiles for any pair of features in the *CelebA* dataset when using the *Left:* VDRL Encoder, and *Right:* DRL Encoder.

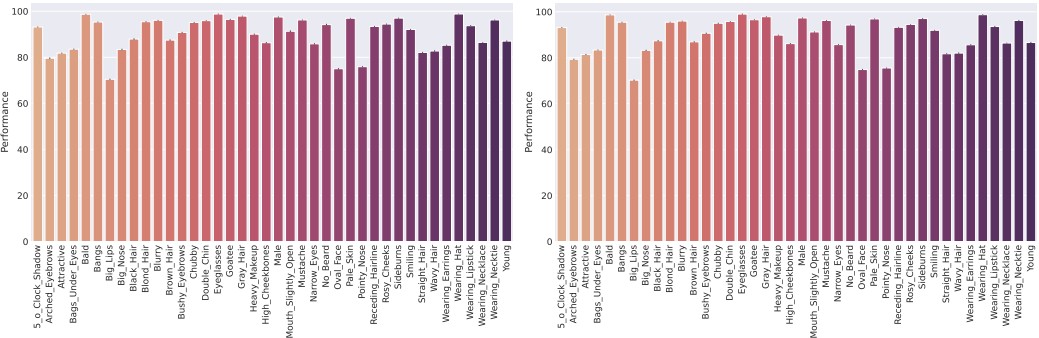

Figure 28: Downstream performance plots for the different features in the *CelebA* dataset when using the *Left:* VDRL Encoder, and *Right:* DRL Encoder.

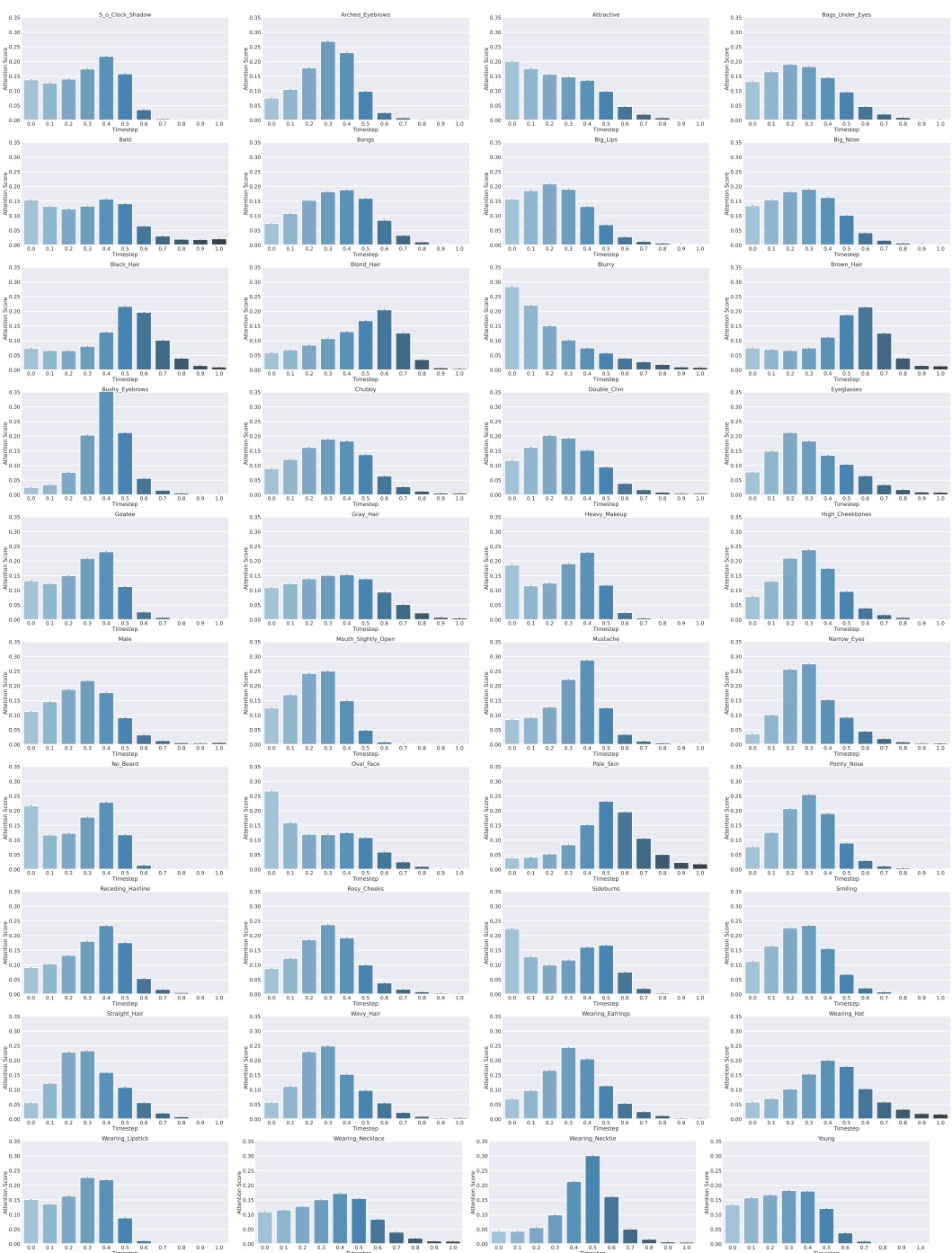

Figure 29: Attention Score profiles for different parts of the trajectory-based representation on CelebA when using the VDRL stochastic encoder.

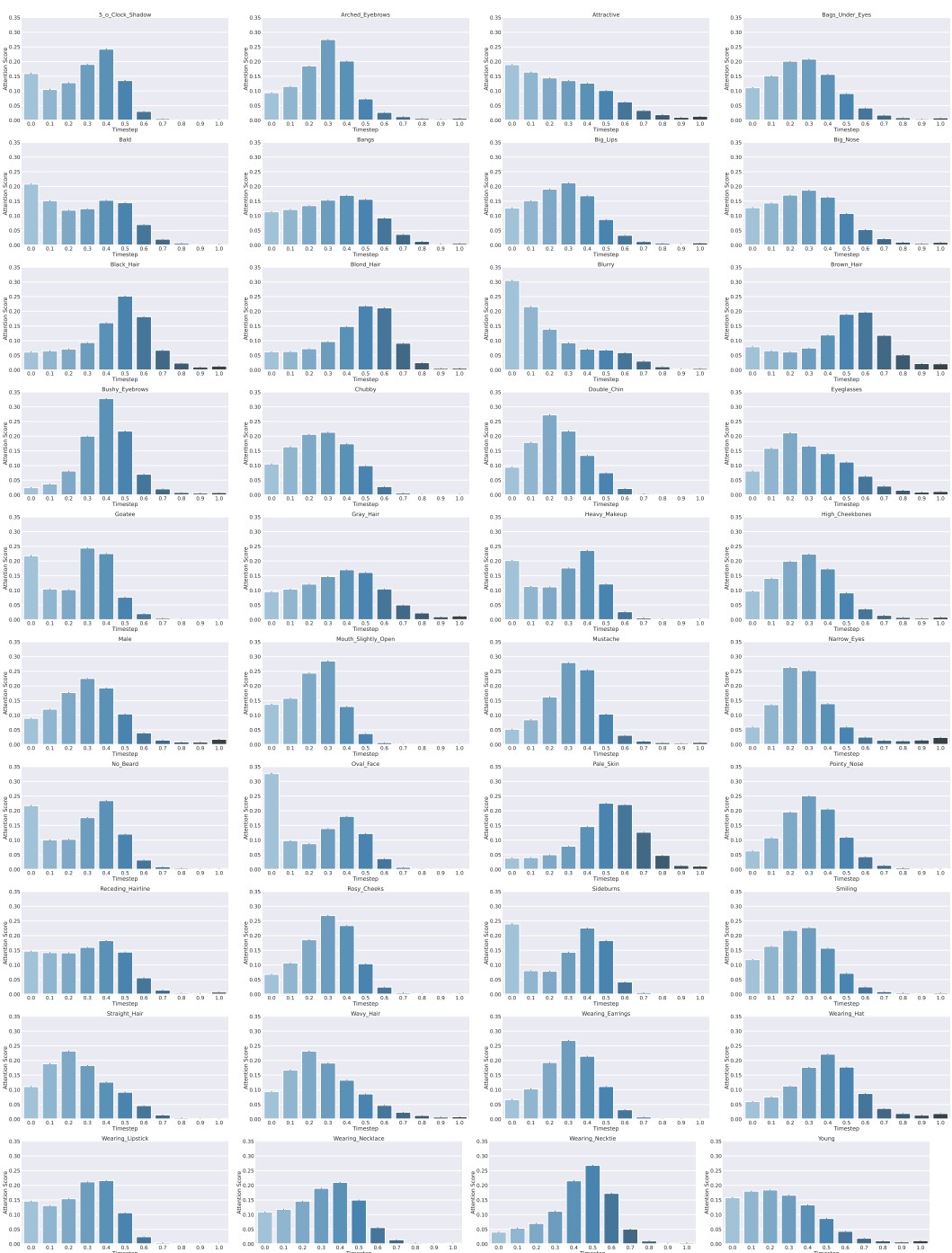

Figure 30: Attention Score profiles for different parts of the trajectory-based representation on CelebA when using the DRL deterministic encoder.

