# OpenReview forum: "From Points to Functions: Infinite-dimensional Representations in Diffusion Models"
_ICLR.cc/2023/Conference — Submitted to ICLR 2023_

### Official Review · Reviewer_WYmT · 2022-10-24

**Confidence:** 3
**Correctness:** 4
**Technical Novelty And Significance:** 2
**Empirical Novelty And Significance:** 2
**Recommendation:** 5

**Clarity, Quality, Novelty And Reproducibility:**

- Clarity (High). The paper is written clearly with many details provided.
- Quality (High). The writing of the paper is of high quality.
- Novelty (Mid-to-low). The core method is from an earlier papers, and the experimental evaluations does not seem to reveal conclusions that are too surprising.
- Originality (High). The work in itself is original as far as I understand.

**Strength And Weaknesses:**

Strength:
- The paper is concise and easy to understand.
- The paper presents extensive empirical studies over the representations, such as the timestep-dependent attention score, mutual information, and downstream application performance.

Weakness:
- Lack of technical novelty: both of the methods DRL and VDRL come from the Abstreiter paper, and it comes not at a surprise that the representations obtained from such models have infinite dimension. The paper mostly presents experimental evaluation results on this type of representations.
- Utility of the representations are not obvious: there are fairly powerful representation learning methods that does not even require a diffusion model. For example, in Figure 1 and Figure 9, the classifier accuracy on CIFAR10 is lower than 80% whereas contrastive learning methods could easily achieve over 90% accuracy with linear classifiers. While the observations are interesting, it is not clear why we want to use diffusion representations for such downstream applications (why not do time-dependent contrastive learning?). Also, there are already works that try to learn diffusion-like methods with powerful representations [1]. All in all, the encoder learned here does not seem very useful empirically.
- Lack of relevance towards existing diffusion models: probably the most interesting part is Figure 5 (c/f), where it suggests that digit identity is highly correlated with large timesteps. However, it is unclear that these findings can be generalized to "standard" diffusion models.

[1] D2C: Diffusion-Denoising Models for Few-shot Conditional Generation, NeurIPS 21.

**Summary Of The Paper:**

This paper discusses representation learning with diffusion models. The key idea is from Abstreiter et al., where a diffusion model is additionally trained on conditioning information given by an encoder. The benefit of this is that is becomes possible to minimize the denoising score matching objective to zero. This paper observes that different representations are learned at different timesteps of the diffusion, and measures the information contained about in the representations, finding that mid-points of the trajectory would give better performance on downstream applications. The paper also investigates other semantic information encoded in these representations.

**Summary Of The Review:**

The paper performs an empirical study over diffusion-based representation learning. While the experiments are extensive, there are not really a lot of technical depth, nor critically useful empirical applications that can be learned from this paper. Even with the entire trajectory of representations, the downstream performance simply does not beat older methods such as contrastive learning. I am also not entirely convinced by this focus on "infinite-size" representations in the title and in parts of the paper -- coming up with "infinite-size" representations is easy (simply add redundancy would work), but such a representation has to be useful for downstream tasks.

---

> ### Author Response · Authors · 2022-11-18
> **Rebuttal**
>
> We thank the reviewer for their valuable comments about our work. We address their concerns below
>
> **Lack of Technical Novelty**: While we agree that we are not introducing a completely new algorithm, we are instead performing novel analysis on an already existing method to describe the benefits that can be obtained from it. Our method of using representations obtained from diffusion models through a transformer is still new and we follow it up with analysis based on the granularity of sampling from the trajectory. Further, we are not aware of any work that tries to explore the kind of information contained at different points in the trajectory, and in general feel that works on analysis should be treated as such and not down-weighted just because they don’t provide new or SoTA results.
>
> **Comparison to other methods**: While Contrastive Learning is a useful paradigm for representation learning, it is important to note that it relies on some notion of invariance in the input domain. For images, this notion can be rotation, translation, etc. but obtaining these invariant manipulations is task-dependent and may not be readily available in some domains, eg. speech. Hence, we feel that representation learning in the absence of these cues, like in Variational Autoencoders or Diffusion-Based Representation Learning is still an important paradigm as it makes no or minimal assumptions about the semantic structure of data and as a result can be applied to novel domains with minimal adjustment or encoding domain-specific invariances.
>
> To answer the reviewer’s question about [1], we note that [1] uses a Variational Autoencoder to learn the representation, and then considers diffusion on top of the representation. It also uses an added contrastive learning loss on the representation. This is different from the approach in consideration as we are not relying on a VAE or reconstruction objective to learn representations, instead it is directly embedded in the diffusion model learning. This is indeed one of the advantages of our method as the representation comes with minimal additional cost when training a generative model.
>
> We are also not clear on what the reviewer means by time-conditioned contrastive learning? What would the time-conditioning impact, since contrastive learning just relies on making two projections of the same image close to each other, while pushing the others further away.
>
> **Focus on “infinite-size”**: While we agree that one could just add redundancy and obtain infinite-dimensional representations, in this work we explicitly show (through different metrics) that that is not the case and that the different parts of these representations are not redundant. Given that the reviewer feels that one could just do this redundantly, we believe that showing that this is actually not the case in these methods has its own merits.
>
> [1] D2C: Diffusion-Denoising Models for Few-shot Conditional Generation, NeurIPS 21.

---

### Official Review · Reviewer_PmAJ · 2022-10-24

**Confidence:** 4
**Correctness:** 3
**Technical Novelty And Significance:** 2
**Empirical Novelty And Significance:** 2
**Recommendation:** 5

**Clarity, Quality, Novelty And Reproducibility:**

**Clarity**

The paper is overall clear. I list the points that are not clear to the readers below:

Equation (5) looks similar to proposition 1 in Abstreiter et al. Is this result from there or derived by the authors? Could the authors provide the corresponding source, e.g., the reference or the proof?

The attention score used in Fig. 3 is not mathematically defined. It is not clear the attention score is calculated with which variables.

The model architecture is not clarified. For example, what is the number of layers of the Wide-ResNet?



**Novelty**

The proposed method has novelty in leveraging the combination of the time-dependent representations, while the method part is extended from previous works like Abstreiter et al.

**Quality**

The paper provides reasonable empirical studies and lacks theoretical analysis. Some important ablations may be in need to improve the paper quality.

**Reproducibility**

The authors include some details for the algorithm and model parameterization, but the provided details are insufficient for reproducing their results.


**Strength And Weaknesses:**

**Strength**

The paper is well-written and organized, making it easy to follow. The claims in the paper are well supported by the corresponding experiments. The authors not only demonstrate the trajectory combination of representations is stronger, but also show which intermediate steps are more important. This observation is important to both diffusion-based model and representation learning community.

**Weakness**

This paper only provides empirical studies on the representations, but lacks theoretical analysis on why the combination of representations is better.

Most of the method part is heavily dependent on Abstreiter et al. and some claims are duplicated, which may weaken the contribution and novelty.

The authors only demonstrate their observations on one single diffusion schedule, while there are various options that may affects the observations. If the diffusion adopts a cosine schedule, can we still get the same conclusions? For example, can we still get the highest attentions scores around t=0.5, as shown in Fig. 3?

On the effects of granularity, the timesteps are only considered to be uniformly partitioned. There could be more combinations to explore. For example, we can sample steps according to a cosine annealing decay weight for the combination.

The accuracy shown in the experiments are relatively lower than the results in both supervised and self-supervised cases. It is hard to say this gap is caused by the method or by using a weaker encoder backbone (I could not find the parameterization details of the Wide-ResNet encoder). It is unclear whether such observation could be helpful in the practical case.

The diffusion schedule is not specified, and some experimental details are not precise; please also see the clarity part.

**Summary Of The Paper:**

The paper presents a trajectory of representations produced by a time-dependent encoder that leverages the diffusion-based method is more informative and can benefit downstream networks (RNN, transformers, etc.) that are able to handle the mixture of the representations. The authors empirically demonstrate their idea on synthetic data and real-world image datasets such as CIFAR-10, mini-ImageNet, colored-MNIST and CelebA.

**Summary Of The Review:**

The topic studied in this paper is important in the study of representation learning and diffusion-based method. However, some key perspectives regarding the proposed method are not comprehensively studied and remain unclear to the readers, which weakens the contribution of this paper. Moreover, the experimental details are not clearly provided. I am inclined to consider this paper below the acceptance threshold given the current version.

---

> ### Author Response · Authors · 2022-11-18
> **Rebuttal (1/2)**
>
> We thank the reviewer for their valuable comments about our work. We address their concerns below
>
> **Theoretical Analysis**: We thank the reviewer for their comments on theory. We also agree that this is an exciting direction of study to better understand diffusion models. As a step toward building such a theory, we believe it is important to identify key mechanisms of interest emerging from these models, such as the distinct and evolving information content we showcase here.
>
> Indeed, we investigate the information content given to the model, and propagated by diffusion trajectories. If the model has been given information about different time-points, it can choose to pick suitable, and even different, time-points to drive differences between two data samples. Thus, a simple way to look at it would be to see that providing the model with all the data can allow the model to figure out on itself where to focus for better class separability. In particular, $\Delta$ is a function that returns separability between two representations (high = more separable), and further suppose you are sampling at $T = \\{t^1, t^2, …, t^N\\}$, then
> $\Delta(E_\phi(x_1, t), E_\phi(x_2, t)) \leq \max_t \Delta(E_\phi(x_1, t), E_\phi(x_2, t))$ for all $t \in T$, and it follows from the definition of max. Further, you can also get $\max_t \Delta(E_\phi(x_1, t), E_\phi(x_2, t)) \leq \max_{t_1, t_2} \Delta(E_\phi(x_1, t_1), E_\phi(x_2, t_2))$ which means you can get better separability if you are allowed to look at different timesteps for different representations. Finally, $\max_{t_1, t_2} \Delta(E_\phi(x_1, t_1), E_\phi(x_2, t_2)) \leq \max_{\psi^1, \psi^2} \Delta(\sum_{i=1}^T \psi^1_{i} E_\phi(x_1, t^i), \sum_{i=1}^T \psi^2_i E_\phi(x_2, t^i))$ showing that we can get even better separability by considering different combinations of representations at different time-steps.
>
> We believe this finding, although not deeply grounded in theory, is valuable in that it articulates key questions that will fuel theoretical analysis from the community. We hope that this insight clarifies the reviewer’s concerns about the proposition as well as theoretical understanding.
>
> **Implementation Details**: We thank the reviewer for putting forward this point. We were planning to open-source our code with the work with embedded documentation and README. We have described the implementation details in Appendix B and further answer the specific questions of the reviewer here.
>
> The implementation requires the score model to be conditioned on a time-dependent encoding of the original sample as well, which means that the network now has additional inputs to consider. Hence picking popular off-the-shelf diffusion models is not as straightforward and one would need to embed its weights in the current system and “fine-tune” it further for representation learning. In this work, we opted for the simpler solution of training the systems from scratch instead. We could use models from Abstreiter et. al directly, but checkpoints for diffusion models with representation learning pipeline are not open-sourced to the best of our knowledge.
>
> We used the NCSN++ model with the Variance Exploding SDE and used the 28x2 WideResNet. In terms of implementation, there will be no change irrespective of which SDE is being used, but the results and attention-profiles might vary. We aim to open-source our code with exact configs to help reproducibility.
>
> **Different Noise Schedules**: We base our work on the continuous-time formulation presented in Yang Song’s Score-based SDE formulation. This formulation can be extended to discrete diffusion models (eg. DDPMs) as well, but then we would not get access to an infinite-dimensional representation and instead, the representation space would be fixed. As we want to look and analyze infinite-dimensional representations, we learn a continuous-time underlying diffusion model which works based on generative and inference SDEs. This prevents us from considering any arbitrary noise schedules and have to instead rely on the SDE formalism, eg. the Variance Exploding SDE, Variance Preserving SDE and finally the sub-VPSDE. In this work we have only looked at the Variance Exploding SDE formulation, and the experiments and implementation details would work out similarly for the other SDE formulations. We cannot do cosine scheduling of noise as we are not aware of an SDE formulation that provides closed form forward distributions at arbitrary time-steps $t$ as well as convergence to some prior.

---

> > ### Author Response · Authors · 2022-11-18
> > **Rebuttal (2/2)**
> >
> > **Classification Performances**: While the performances seem lower than supervised and self-supervised domains, we would like to stress that this work considers a completely unsupervised domain. That being said, our goal was not to attain SOTA and we did not try  very large architectures or custom-designed losses, instead sticking to the standard ScoreSDE formulation without additional bells and whistles. This was a conscious choice in order to focus on the information content of diffusion trajectories with as little confounders as possible.
> >
> > In theory, we believe that one can boost performance of these systems as well through more compute and tuning, but that is tangential to our main message of analysis on the infinite dimensional object.
> >
> > **Attention Scores**. Given a set of representations $\\{z_i\\}$ with $i=1,.., T$ for any given data sample, we concatenate another embedding vector $e$ to it and then feed it to the transformer, and finally take the output corresponding to $e$. In case of attention profile analysis, we use a single layer transformer, and visualize the attention score between a query coming from $e$ and keys coming from $z_i$, averaged over the different number of heads. This, on average, provides the relevancy of each time-step.

---

### Official Review · Reviewer_ZUwW · 2022-10-25

**Confidence:** 4
**Correctness:** 3
**Technical Novelty And Significance:** 2
**Empirical Novelty And Significance:** 2
**Recommendation:** 3

**Clarity, Quality, Novelty And Reproducibility:**

Clarity: the paper is clear.

Quality: the quality of the paper is limited. See details in the weaknesses above.

Novelty: the novelty is limited compared to the prior work (Abstreiter et al 2021).

Reproducibility: it seems reproducible.

**Strength And Weaknesses:**

**Strength**
1. The paper is clearly written and well organized.

**Weaknesses**
1. The paper is incremental compared to the prior work (Abstreiter et al 2021). In fact, it is exactly the same to the prior work except that it considers to leverage the "noisy" representations along the trajactory simultaneously for downstream tasks.

2. I think the claim of "points to functions" and "infinite dimensional representations" should be modified, including the title. This is very misleading because some popular algorithms in machine learning like Kernel methods can really do this in function space. However, The proposed method (using RNNs) takes representations at finite time steps as input instead of the whole function.

3. I did not see a clear advantage of diffusion models to learn representations. In fact, the paper does not compare to any other deep generative models like VAEs and GANs or discuss about this. It it not well motivated for me.

**Summary Of The Paper:**

The paper studies the representation learned by diffusion models equipped with an encoder. Based on the prior work (Abstreiter et al 2021), the paper obtained a time-dependent encoder along with the training of a diffusion model.

In contrast to the prior work (Abstreiter et al 2021), which uses the output of the encoder at a single timestep (i.e. t0), the paper considers the discretized trajectory at multiple timesteps simultaneously. It is done by using an RNN or a transformer to aggregate the sequential representations together and make a final decision.

Empirically, it investigates the benefits of the trajectory information for downstream tasks, analyzes the different information encoded in the trajectory in different time steps by mutual information, parses the semantic information along the trajectory and demonstrates the benefits of using more samples in the trajectory.


**Summary Of The Review:**

This paper is clearly below the acceptance bar of ICLR due to the lack of novelty and motivation.

---

> ### Author Response · Authors · 2022-11-18
> **Rebuttal**
>
> We thank the reviewer for their valuable comments about our work. We address their concerns below
>
> **Novelty**: The similarity to Abstreiter et. al appears because we are performing analysis on the method that was introduced in Abstreiter et. al. We kindly reiterate that we are not claiming a  novel representation learning methodology but instead are focusing on the  benefits of the infinite representation obtained from such a method. While Abstreiter et. al proposes a representation learning paradigm capable of learning an infinite-dimensional representation, it doesn’t comment on the kind of information stored in different parts of this object, whether there are redundancies or whether there is any need for learning an infinite-dimensional object. Our aim is to answer precisely these questions. We analyze, through attention profiles and mutual information, that different parts of this trajectory actually encode different information, and show its benefits for downstream prediction (eg. the transformer model over multiple points)
>
> **Claim on Infinite-Dimensional Representation**: The proposed method of using RNN or Transformer on the representations learned does indeed discretize the trajectory, however it still remains the case that the trajectory itself is continuous and in that sense, infinite-dimensional. There is no restriction on how finely one can discretize this trajectory. In fact, we do a thorough analysis on the granularity of this discretization. Note that this is similar in spirit to other infinite-dimensional and continuous systems. For example, Neural ODEs [1] are continuous-time systems but they still rely on an ODE solver which, at the base of it, does perform discrete operations. Similar things can be said about bayesian non-parametric systems where they indeed have the capacity to learn an unbounded number of parameters but conditioned on the data, still only learn a finite number  Nevertheless, we acknowledge that the use of “infinite-dimensional” may be misleading so we clarify it’s context in the revised manuscript in the Introduction.
>
> **Comparison to other Representation Learning Methods**: We refer the reader to the original Diffusion-Based Representation learning work by Abstreiter et. al for comparisons against other representation learning methods like VAEs and SimCLR. Our aim in this work was to provide more insights into the workings and content of infinite-dimensional representations obtained, conditioned on the fact that they outperform VAE style systems for representation learning as described in Abstreiter et. al.

---

> > ### Comment · Reviewer_ZUwW · 2022-11-19
> > **Acknowledgement to the rebuttal**
> >
> > I appreciate the author's rebuttal. Unfortunately, I still think it is incremental work and the "infinite-dimensional" claim is very misleading. Therefore, I keep my score. I explain more about my opinion on the "infinite-dimensional" issue.
> >
> > In fact, It is not exactly the same as the Neural ODE case. Neural ODE indeed learns a potentially "infinite-dimensional (or layer)" model and the solver can discretize it with different time schedules. In this case, the **prior work instead of this paper learns "infinite-dimensional" features** but the **RNN (the method of this paper) only takes finite sets of features as input**. If any downstream task requires a different number of features or different time schedules, the proposed method should be retrained.

---

> > > ### Author Response · Authors · 2022-11-25
> > > **Response**
> > >
> > > We thank the reviewer for their response; and agree that the methods provided for downstream classification (eg. RNNs/Transformers) are not continuous-time or infinite dimensional and we haven't claimed that as well. Our claim is just that the **representations** obtained from Diffusion-Based Representation Learning are infinite-dimensional, which we hope that the reviewer agrees with. Yes, our downstream analysis and classification relies on discretization of the trajectory representation, but the underlying object is still infinite-dimensional and we believe that discretized analysis of this representation reveals key insights on what features are encoded at different points, irrespective of whether the underlying object is discrete or continuous.

---

### Official Review · Reviewer_tLzz · 2022-10-25

**Confidence:** 3
**Correctness:** 3
**Technical Novelty And Significance:** 2
**Empirical Novelty And Significance:** 3
**Recommendation:** 3

**Clarity, Quality, Novelty And Reproducibility:**

The paper is clearly written. Certain parts of the experimental evaluation should be further clarified (see above). The proposed idea is original but incremental to the diffusion-based representation learning paper.

**Strength And Weaknesses:**

Strengths:

* The idea of the paper is simple, yet effective.
* The paper is well-written.
* The authors experiment in a wide variety of settings, from synthetic tasks to more realistic tasks, such as CelebA classification.
* The authors demonstrate (using the attention profiles and the estimated Mutual Information) that the representations of the image are changing over the diffusion time.

Weaknesses:

* The paper does not compare with other methods of representation learning, e.g. Contrastive Learning. Given that the diffusion-based representation learning paper itself has not yet been published, I think it would be useful to compare this new work to more established methods for learning representations.
* There is a lack of clarity for some things regarding the implementation. Why did the authors need to retrain the models from Abstreiter et. al? Aren't there open-source models available that could have been reused? If not, which models did the authors train? Is this the NCSN++ model with the Variance Exploding SDE (as it is implied from the Background Section)? Would anything change if instead of the Variance Exploding SDE, one used Variance Preserving or the sub-VP SDE?
* In the experiments of Figure 1 (but also in subsequent experiments), it seems that we are comparing a Transformer that takes as input the whole trajectory vs an MLP that takes as input a single point. Does the transformer and the MLP have the same number of parameters? If the MLP has much less expressive power than the Transformer, it could just be that the observed benefits over the baseline are due to bigger or more powerful architecture (and not because of the trajectory vs single point).
* It would have been more impactful to report numbers on more standard benchmarks such as CIFAR10 , CIFAR100 or ImageNet. This would give a better sense of how this method compares to other methods for learning representations (e.g. SimCLR).

**Summary Of The Paper:**

The paper proposes a new way to solve downstream tasks using diffusion-based representation learning. The authors observe that representation learning inspired by diffusion doesn't give one, but infinitely many representations of the image (one for each time, t, of the diffusion). The work that introduced diffusion-based representation learning used one of those representations to optimize for a downstream task. Instead, this paper proposes to consider the whole trajectory of representations. Particularly, the authors propose to use a Transformer (or an RNN) to map from the trajectory to a single embedding that is then used to solve downstream tasks. Experimentally, this yields improved performance in a wide variety of settings.

**Summary Of The Review:**

The paper proposes a better way to solve downstream problems with diffusion-based learned representations. The key innovation is to use the whole trajectory and not a single point to extract a representation that will be used to solve downstream tasks. The method is not evaluated against other ways to learn representations and hence its practical relevance is not clear.

---

> ### Author Response · Authors · 2022-11-18
> **Rebuttal**
>
> We thank the reviewer for their valuable comments about our work. We address their concerns below
>
> **Comparison to Contrastive Learning**: We refer the reviewer to the original Diffusion-Based Representation Learning (DRL) work where they provide a comparison with SimCLR or other representation learning baselines. Our aim for this work is not to set SoTA on classification based tasks but instead to perform analysis on the nature of the representations learned in the DRL framework to understand whether learning of this infinite-dimensional object is indeed meaningful, if it encodes different information at different times, or if it is simply redundant and one can randomly pick any. Beyond classification, diffusion trajectories are under active investigation for manipulation and interventions for data editing purposes [1].
>
> As a paramount example, we found a strong bidirectional connection between our work and SDEdit [1] in the following sense:
> The fact that SDEdit works partially confirms our empirical results.
> Our work gives a direction for how to improve SDEdit.
> We added an extensive account of this connection in the Appendix of the revised manuscript.
>
>  A better understanding of diffusion-supported representations is therefore of interest to a wide community.
>
> **Implementation Details**: We thank the reviewer for putting forward this point. We were planning to open-source our code with the work and have implementation details in Appendix Section B, where we have added some additional details. We also answer the specific questions of the reviewer here.
>
> The implementation requires the score model to be conditioned on a time-dependent encoding of the original sample as well, which means that the network now has additional inputs to consider. Hence picking popular off-the-shelf diffusion models is not as straightforward and one would need to embed its weights in the current system and “fine-tune” it further for representation learning. In this work, we opted for the simpler solution of training the systems from scratch instead. We could use models from Abstreiter et. al directly, but checkpoints for diffusion models with representation learning pipeline are not open-sourced to the best of our knowledge.
>
> We used the NCSN++ model with the Variance Exploding SDE. In terms of implementation, there will be no change irrespective of which SDE is being used, but the results and attention-profiles might vary. We aim to open-source our code with exact configs to help reproducibility.
>
> **Expressive Power**: We make sure that the trend that we see is not because of the computational budget of the models as we tested transformer architectures with just single-point inputs, and these gave similar performances as the MLP systems while maintaining the same expressive power as the multi time-step transformer experiments. Hence we can conclude that the trends we see are not an outcome of model size.
>
> **Numbers on CIFAR10/CIFAR100/Mini-ImageNet**: We refer the reviewer to Figure 1 which contains performance results for CIFAR-10, CIFAR-100 and Mini-ImageNet. Was the reviewer looking for some other metrics concerning these datasets? We would be happy to discuss these further given specifics.
>
> [1] Meng, C., Song, Y., Song, J., Wu, J., Zhu, J.Y. and Ermon, S., 2021. Sdedit: Image synthesis and editing with stochastic differential equations. arXiv preprint arXiv:2108.01073.

---

> > ### Comment · Reviewer_WYmT · 2022-11-18
> > **Questions about SDEdit**
> >
> > I am curious about the claims on SDEdit, as it is a widely applied method, e.g., in Stable Diffusion. However, I have several questions over the proposed connection between diffusion representation learning and SDEdit:
> >
> > - Since all the initialization are Gaussians with the same mean, the mixing is simply just the Gaussian with the same mean but a different variance?
> > - Can you propose an algorithm based on your proposed ideas and validate the claims that "mixing should be a more suitable starting point"? I could not imagine how to implement SDEdit with this --- what noise level should you choose?
> > - Can you use your method to decide what is the best hyperparameter for SDEdit?

---

> > > ### Author Response · Authors · 2022-11-19
> > > **Connections with SDEdit**
> > >
> > > Thank you for the acute comments about the described connection between our work and SDEdit. We answer each of your questions in the followings:
> > >
> > > **Since all the initialization are Gaussians with the same mean, the mixing is simply just the Gaussian with the same mean but a different variance? Can you propose an algorithm based on your proposed ideas and validate the claims that "mixing should be a more suitable starting point"? I could not imagine how to implement SDEdit with this --- what noise level should you choose?**
> > >
> > > It is true that the weighted sum of independent Gaussians will be Gaussian as well and that was the reason we used "mixing" instead of "mixture" in the text. As in probability and statistics, mixture implies a linear weighted some (e.g. Mixture of Gaussians), we intended a more general information aggregation by the word "mixing" instead of "mixture". Obviously, we cannot simply concatenate $x^{(g)}(t_j), j\in J$ as the dimension of the initial state of the backward diffusion process has to be the same as the data space. However, a potential candidate would be a trainable nonlinear mixer that takes guides at different times and learns to mix them in an end-to-end way.
> > >
> > > **Can you use your method to decide what is the best hyperparameter for SDEdit?**
> > >
> > > It is observed in our empirical results e.g. (Figures 3, 4, 5) that the weight distribution assigned to time steps by the attention model is mostly single modal and in some cases highly peaked. It seems logical to assume the time step that received the larger attention weight should be chosen as $t_0$ for the SDEdit algorithm.
> > >
> > > We hope this could answer your questions. Of course, these points have to be validated in experiments on SDEdit which can indeed be an interesting application of our work in the future. Please let us know if we can clarify any other questions in this regard.

---

### Decision · Program_Chairs · 2023-01-20

**Decision:**

Reject

**Justification For Why Not Higher Score:**

 The paper ultimately appears rather incremental, both in the quality of the results and methodologically from the previous Abstreiter et al., 2021 paper.

**Justification For Why Not Lower Score:**

 NA

**Metareview: Summary, Strengths And Weaknesses:**

Thank you for your submission to ICLR.  The reviewers and I are in agreement that although there are some interesting aspects to the proposed work, ultimately the paper seemed rather incremental relatively to the Abstreiter et al., 2021 paper: specifically the main contribution of this paper seems to be that the representation learning of this past work can perform better if a mix of information along the entire trajectories is used instead of just the final state.  This is an interesting insight, but seems to be more suitable to a technical note or part of a larger whole paper (for example, if similar approaches could be applied to other representation learning settings), rather than an independent paper by itself.  Thus, the consensus of the reviewers was that this paper wasn't suitable for ICLR in its current form.